# Chamber-specific chromatin architecture guides functional interpretation of disease-associated *Cis*-regulatory elements in human cardiomyocytes

S. Haydar [1,2], R. Bednarz[1,2,3], P. Laurette [1,2,3], I. Sobitov [4,5,6],
N. Díaz i Pedrosa[1,2,3], P. Videm[7], T. Lueneburg[1], S. Kuß[1], H. Lahm[8], M. Dreßen [8],
M. Krane[8,9], C. Schmidt [6,10,11], B. A. Grüning [7], N. Voigt [4,5,6],
K. Streckfuss-Bömeke [6,11,12] & R. Gilsbach [1,2,3,13] ✉

*Cis*-regulatory elements (CREs) are noncoding DNA regions regulating cell-type-specific gene expression programs by interacting with distal gene promoters. Here, we aim to decode the function and spatial organization of CRE-promoter interactions in human cardiomyocytes. We analyzed the epigenome and chromatin interactions of human male atrial, ventricular, and failing cardiomyocytes. Atrial and ventricular cardiomyocytes harbored chamber-specific CRE-promoter interactions modulating gene expression as confirmed by functional epigenetic silencing. These CRE-promoter interactions explain the distinct contribution of non-coding genetic variants to atrial and ventricular diseases, such as dilated cardiomyopathy and arrhythmias. We dissected the prototypic *KCNJ2* locus, encoding a potassium channel associated with ventricular arrhythmia susceptibility. Functional epigenetic silencing confirmed that CREs, harboring QT-duration-associated genetic risk factors, modulate *KCNJ2* gene expression levels, alter KCNJ2-dependent channel currents, and affect cardiomyocyte repolarization. The presented human CM-specific chromatin interaction analysis provides key insights into regulatory mechanisms and aids in interpreting genetic risk factors.

Cardiovascular diseases are among the leading causes of death in Western countries[1]. Heart diseases are driven by altered gene expression programs. Gene expression is controlled by distal *cis*-regulatory elements (CREs)[2–5] in a time- and cell type-specific manner. The human genome harbors more than 1 million CREs[4,5]. These CREs are established and bound by transcription factors (TFs) and interact with target genes to modulate their expression[6]. CRE-target gene interactions can span large genomic distances but predominantly occur within topologically associating domains (TADs). TADs are regulatory domains formed through a loop extrusion process where ring-shaped cohesin complexes extrude a DNA loop until a boundary element is halted, often bound by the CCCTC-binding factor (CTCF)[7]. This dynamic process brings CREs and associated transcriptional regulators in frequent spatial proximity to their targets while insulating them from genes outside of the TAD. Sub-megabase-sized TADs form higher-order chromatin compartments, segregating the genome into active (A) and inactive (B) chromatin compartments[8].

Epigenomic profiling methods allow annotation of putative CREs in an essentially unbiased manner. CREs are decorated by specific histone modifications (like H3K4me1), high chromatin accessibility,

and low DNA methylation (LMR). These features are dynamically established during heart development and can be altered in disease[9–15]. The establishment of active CREs (enhancers) decorated by H3K27ac has been studied in fetal, adult, and diseased human hearts using cardiac tissue[14,16,17] and human CMs[11]. A comparative analysis of mouse heart tissue and different cardiac cell types, including CMs, showed that CREs are established and respond to heart disease in a cell-type-specific manner[18]. A joint analysis of histone marks and DNA methylation in human CMs identified more than 100,000 CREs, many of which are enriched for non-coding variants associated with heart disease[11,14,19]. Thus, genetic variants within CREs may exert cell-type-specific effects; for example, ischemic heart diseases are likely linked to CRE variants in endothelial cells[20], while cardiac arrhythmia and hypertrophy are likely associated with CRE variants in CMs. Such links may even differ between distinct CM populations (e.g., atrial, ventricular, or conduction system CMs)[21].

Non-coding variants can modify TF binding to CREs, which can subsequently promote or disrupt the interaction dynamics of CREs with their target gene promoter, thereby leading to changes in gene expression[22]. Therefore, interpreting non-coding variants in the light of cell-type-specific CREs and chromatin interaction maps appears essential. High-throughput chromosome conformation capture sequencing (Hi-C) provides genome-wide resolution of chromatin interactions[23], unlike targeted methods such as Hi-ChIP[24] and Capture-Hi-C[25] that rely on post-translational modifications or predefined probes. Notably, Hi-ChIP's reliance on post-translational modifications limits its detection sensitivity[26]—unmodified regions are not assessed – and makes it challenging to distinguish true chromatin rewiring events from dynamic changes in chromatin states, thereby complicating mechanistic interpretation.

In this study, we aimed to resolve distal interactions of CREs with target promoters in human cardiomyocytes to link them to gene expression programs and genetic risk factors and to show their functional relevance through targeted functional perturbation. Therefore, we performed high-resolution Hi-C analysis of healthy atrial and ventricular CMs, failing ventricular CMs, and ventricular tissue to capture interactions independently of the epigenetic landscape. We integrated these data with epigenetic features and gene expression profiles. A comparative analysis demonstrated that CM-specific chromatin interaction data markedly outperform cardiac tissue-level data for sensitive detection of CM promoter interactions.

CM-specific data revealed distinct chromatin interaction landscapes between atrial and ventricular CMs. These chamber-specific interactions aligned with differential gene expression, including chamber-specific TFs, and were enriched for genetic risk variants associated with atrial and ventricular diseases, respectively. In ventricular CMs of failing hearts, promoter rewiring was linked to pathological gene expression of only eight genes and had no impact on genetic risk factor linkage. Using CRISPR interference (CRISPRi)-mediated perturbation[27], we validated the functional relevance of several disease-associated chromatin interactions. Decoding of the distal regulatory landscape of the *KCNJ2* locus identified regulatory elements controlling *KCNJ2* expression, hence modulating KCNJ2-dependent repolarization currents.

Together, our findings advance the understanding of the regulatory architecture underlying CM function and support the interpretation of non-coding variants, providing a framework for prioritization of genome-wide association studies (GWAS) signals in human CM.

## Results

### A comparative analysis of chromatin interactions in human cardiac tissue and cardiomyocyte nuclei

CREs can be annotated using epigenetic signatures. Previous studies have shown that these signatures are highly cell-type-specific in the

heart[18,19,28]. We therefore investigated whether CM-specific analysis of chromatin interactions could provide insights beyond those obtained from bulk cardiac tissue nuclei. To this end, we isolated bulk cardiac tissue nuclei or CM nuclei from three non-failing left ventricles (NF-LV) using fluorescence-activated nuclei sorting (FANS). Bulk tissue nuclei (NF-LV-tissue) were identified using the DNA dye Draq7, while CM nuclei (NF-LV-CM) were further specifically labeled with antibodies targeting pericentriolar material 1 (PCM1) and phospholamban (PLN)[11,13,29] (Fig. 1a, b). Hi-C data of bulk cardiac nuclei and CM nuclei were generated from matched biological samples using the same methodology to avoid biases and minimize technical variability (Supplementary Fig. 1). We generated a chromatin state model for CM and cardiac tissue using published ChIP-seq data[11] (Supplementary Fig. 2a) using ChromHMM[11,30,31]. *Cis*-regulatory elements (CREs) were identified as regions with low CpG methylation (LMRs)[11,30,31] (Supplementary Fig. 2b). As an example, the locus containing the cardiac alpha-actin (*ACTC1*) gene is shown in Fig. 2a. The Hi-C contact map and virtual viewpoint analysis indicate lower interaction frequencies between the *ACTC1* promoter and CREs in NF-LV-tissue compared to NF-LV-CM (Fig. 2a). We next expanded this analysis genome-wide. A comparative viewpoint analysis of more than one hundred CM marker genes[32] revealed that CM-specific Hi-C data identify twice as many promoter-interacting domains (PIDs) in CM-specific data compared to tissue-level data, with a higher cumulative domain size (2.2 fold) and relative interaction frequency (1.98 fold) (Fig. 2b–d; Supplementary Figs. 3–7; Supplementary Data 1). In contrast, no significant differences were observed for housekeeping genes stably expressed across cardiac cell types[18] (Fig. 2b–d) while several endothelial-, fibroblast-, and smooth muscle cell gene loci showed significantly higher interaction frequencies in NF-LV-tissue compared to NF-LV-CM (Supplementary Figs. 8–10). To determine whether these differences reflect changes in higher-order chromatin organization, we performed a principal component analysis (PCA) of the Hi-C contact matrices to infer A/B compartments. We identified 132 domains (covering 53.6 Mb, ~1.9% of autosomal genome) that switched compartment status between CM and tissue nuclei (Supplementary Fig. 11a). Notably, several CM-enriched genes including the muscarinic receptor M2 (*CHRM2*), sarcoglycan delta (*SGCD*), and the ryanodine receptor 2 (*RYR2*) resided in active (A) compartment in CM but inactive (B) compartment in bulk cardiac tissue nuclei (Supplementary Fig. 11b–d), highlighting the cell-type specific nature of higher-order chromatin structure.

Together, these results demonstrate that chromatin interactions and 3D genome organization in the human heart are highly cell-type-specific. CM-resolved analysis is therefore critical for accurately decoding the regulatory architecture of CMs.

### Generation of human CM-specific Hi-C data for differential chromatin interaction analysis

The next aim was to generate CM-specific contact maps from a total of 23 biological replicates of non-failing left atria and left ventricle (NF-LA-CM, NF-LV-CM) as well as terminally failing ventricles (F-LV-CM; ejection fraction [EF] <25%, Fig. 1c), for differential chromatin interaction analysis. In total, we generated Hi-C data from 9 NF-LV-CM, 6 NF-LA-CM, and 8 F-LV-CM samples with a total sequencing depth of 8.6 billion reads. Demographic characteristics were comparable across groups (Fig. 1c, Supplementary Data 2). Each biological replicate was sequenced to a depth of at least 200 million reads (Supplementary Fig. 1a, Supplementary Data 3). Quality control metrics demonstrated a high within-group correlation of chromatin interactions (Supplementary Fig. 1b), and the distribution of short- and long-range contacts was similar across all samples (Supplementary Fig. 1c).

### Higher-order genome organization is conserved in human CM

We next asked whether higher-order chromatin organization, including TADs and A/B compartments, is different between the studied

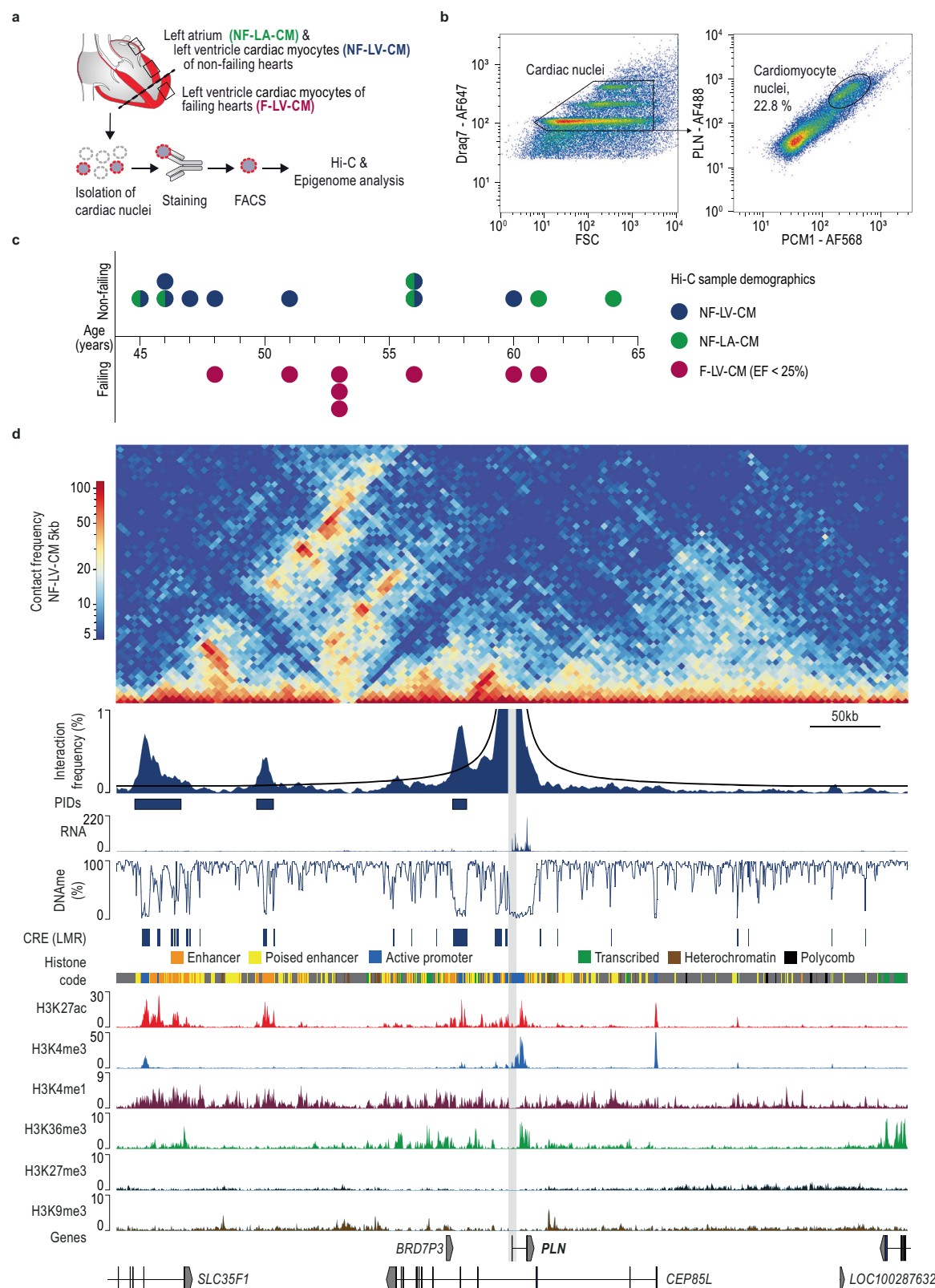

groups. Analysis of TADs revealed no significant differences among the different CM groups, since TAD insulation scores were highly correlated, and boundary strengths remained preserved across conditions (Supplementary Fig. 11e–f).

Similarly, principal component analysis of the Hi-C contact matrices revealed minimal differences in chromatin compartmentalization among groups. A comparison between F-LV-CM and NF-LV-CM

identified only 18 genomic regions with A/B compartment switches (~3 Mb, ~0.1% of the autosomal genome) (Supplementary Fig. 11a). Notably, A/B compartment switches in failing CM were not associated with altered epigenetic marks or gene expression. Likewise, comparison of NF-LA-CM and NF-LV-CM identified only 7 switching A/B domains (~3.2 Mb, ~0.1% of the autosomal genome, Supplementary Fig. 11a), including loci harboring *ACTA2* and *PITX2*.

**Fig. 1 | Chromatin interaction and epigenome analysis of human cardiomyocyte nuclei. a** Workflow for the isolation of human cardiomyocyte (CM) nuclei derived from adult non-failing human left ventricle (NF-LV-CM), non-failing left atrium (NF-LA-CM), and failing left ventricle (F-LV-CM) for epigenetic and chromatin interaction analysis. **b** A representative fluorescence-activated nuclei sorting (FANS) analysis of cardiac nuclei stained with Draq7, anti-pericentriolar material 1 (PCM1) and anti-phospholamban (PLN) antibodies. 5000 events are displayed, and the gating strategy to isolate CM with high purity is highlighted. **c** Biometric data of human samples used for high-throughput chromosome conformation capture (Hi-C). EF, ejection fraction. **d** Original traces of chromatin interaction frequency heatmap at 5 kb resolution (Hi-C), virtual viewpoint analysis of *PLN* promoter interactions (1 kb resolution), gene expression (RNA-seq), mCpG (DNA methylation sequencing), histone modification ChIP-seq and histone code (ChromHMM). Low-methylated regions (LMR) and PIDs are annotated. Gray area highlights the *PLN* promoter. Hi-C data is merged from 9 biological replicates. CRE, *cis*-regulatory elements.

Collectively, these results indicate that both TAD structure and A/B compartmentalization are highly conserved across CM subtypes despite associated transcriptional and epigenetic changes.

## CM-specific promoter interaction analysis identifies spatially interacting CREs

Interactions of target gene promoters primarily occur within higher-order chromatin structures, such as TADs and A/B compartments. To resolve these interacting regions, we performed a genome-wide virtual viewpoint analysis for all coding-gene promoters. To characterize these interactions, we further integrated chromatin state annotations, DNA methylation profiles, and nuclear RNA-seq data, as shown in Fig. 1d for phospholamban (*PLN*). We used a DNA methylation-guided annotation for unbiased annotation of CREs across all groups (Supplementary Fig. 12). Consistent with established regulatory features, these CREs were enriched for H3K4me1, H3K27ac, and accessible chromatin, validating the annotation strategy[6,30,31]. Over 50% of annotated CREs overlapped with enhancer elements (H3K27ac+, H3K4me1+) and 20% with poised enhancer elements (H3K4me1+) (Supplementary Fig. 2c, d). In NF-LV-CM, we detected 357,918 PIDs spanning 54 Mb, with a median length of 7 kb (Supplementary Fig. 13a, b, Supplementary Data 4). A median number of 5 PIDs was detected per gene, located at a median distance of 147 kb from the promoter (Supplementary Fig. 13c, d). The majority of PI-CREs (77%) spanned one to multiple reference gene promoters (Supplementary Fig. 13e). PIDs overlapped 26 Mb of CRE sequence, and 9.8 Mb of ChromHMM-annotated[33] enhancers (Supplementary Fig. 13a, f). Promoter-interacting CREs (PI-CRE) accounted for 36.8% of all annotated CREs and 41.5% of total CRE sequence length (Supplementary Fig. 13f). Based on ChromHMM classification, 54% of the promoter non-interacting CREs (PN-CRE) and 65% of the PI-CREs were classified as enhancers (Supplementary Fig. 13g). Notably, PI-CREs exhibited higher levels of H3K27ac marking compared to PN-CREs, suggesting enhanced regulatory activity (Supplementary Fig. 13h). Interestingly, across the NF-LV-CM, NF-LA-CM and F-LV-CM PIDs (Supplementary Data 4–6) we consistently identified between 55,000 and 59,000 PI-CREs per group (Supplementary Data 7–9).

## CRISPRi-mediated silencing validates the functional relevance of PI-CREs

To assess the functional relevance of PI-CREs, we employed CRISPRi, which enables targeted epigenetic silencing without altering the genomic sequence. CRISPRi relies on the RNA-guided recruitment of a catalytically inactive Cas9 (dCas9) fused to a KRAB repressor domain to a specific genetic locus. CRISPRi vectors can be effectively delivered to human induced pluripotent stem cell-derived CMs (hiPSC-CM) using lentiviruses or adeno-associated viruses (AAVs) (CRISPRi[Lenti], CRISPRi[AAV])[27,34,35]. We previously demonstrated that AAVi robustly represses target gene promoters in CMs, inducing local heterochromatin formation marked by specific H3K9me3 deposition and loss of chromatin accessibility[34].

To evaluate the utility of CRISPRi for the functional interrogation of PI-CREs, we targeted several cardiac enhancers in ventricular hiPSC-CM (hiPSC-vCM), which were previously ablated. These experiments confirmed the functional link of two enhancers proximal to the *MYL2* and *MYH7* transcriptional start sites, which had previously been shown to control gene expression in mice and human embryonic stem cell-derived CM (ES-CM)[9,36]. We further validated an intronic *KCNH2* enhancer reported by Hocker et al.[19] as well as an *SCN5A* super-enhancer domain[37] (Supplementary Figs. 14a–c, 15), both genetically linked to cardiac arrhythmias[19,37]. Silencing an alternative *SCN5A* super-enhancer domain did not impact *SCN5A* gene expression, consistent with prior genetic deletion experiments in mice[37], reinforcing the validity of our CRISPRi approach and highlighting the functional conservation of enhancers across mammalian species.

Importantly, while genomic deletion of regulatory loci can lead to unintended chromatin rewiring[38] and genomic rearrangements[39], CRISPRi-mediated silencing does not modify the genome. Silencing of four loci did not impact the interaction strength with their linked promoter (Supplementary Fig. 14d–f). This suggests that the observed transcriptional effects are due to enhancer silencing rather than to disruption of 3D chromatin architecture. Together, these findings establish CRISPRi as a robust tool for the functional validation of PI-CREs in hiPSC-CM.

## Chamber-specific promoter-CRE interactions are linked to gene expression

While atrial and ventricular CM share most of their gene expression repertoire, they differ in the relative abundance of transcripts[40]. To investigate whether these differences are associated with chamber-specific chromatin interactions, we analyzed Hi-C data of NF-LV-CM and NF-LA-CM (Supplementary Data 4 and 5).

Using a replicate-based analysis, we found 1,446 differential PIDs (Fig. 3a), of which 62% were characteristic of LA and 38% of LV. Chamber-specific PIDs were correlated with chamber-specific gene expression (Fig. 3b) and functional enrichment analysis linked them to pathways involved in CM differentiation and cardiac development processes (Fig. 3c). This includes *ADRA1A, CCN2, HEY2*, and *TBX5*; the latter two being key transcription factors known to drive chamber-specific gene expression[41] (Fig. 4a, b, Supplementary Figs. 16, 17).

Next, we asked whether CRISPRi-mediated silencing of chamber-specific PI-CREs in hiPSC-vCM or hiPSC-aCM (atrial hiPSC-CM) can show their functional relevance. We first tested whether the hiPSC-aCM and hiPSC-vCM display the expected gene expression patterns using RNA-seq. These data confirmed that hiPSC-aCM and hiPSC-vCM expressed the expected marker genes and key chamber-specific genes studied in this project (Supplementary Fig. 18).

Silencing of PI-CREs linked to *HEY2* and *TBX5* significantly reduced the expression of their respective target genes, confirming the regulatory function of these elements (Fig. 4c, d).

We integrated data from the VISTA enhancer database[42] to investigate whether chamber-specific gene expression can be driven by CREs independently of chromatin interactions. Of 380 enhancers shown to be active in the fetal mouse heart, 43% did not overlap with any PID in adult NF-LA-CM or NF-LV-CM (Fig. 3d). GREAT analysis[43] revealed that these non-promoter-interacting VISTA enhancers were located within regulatory domains of genes involved in general and early development processes (Fig. 3e), suggesting that their activity is restricted to embryonic stages studied in the VISTA project or specific

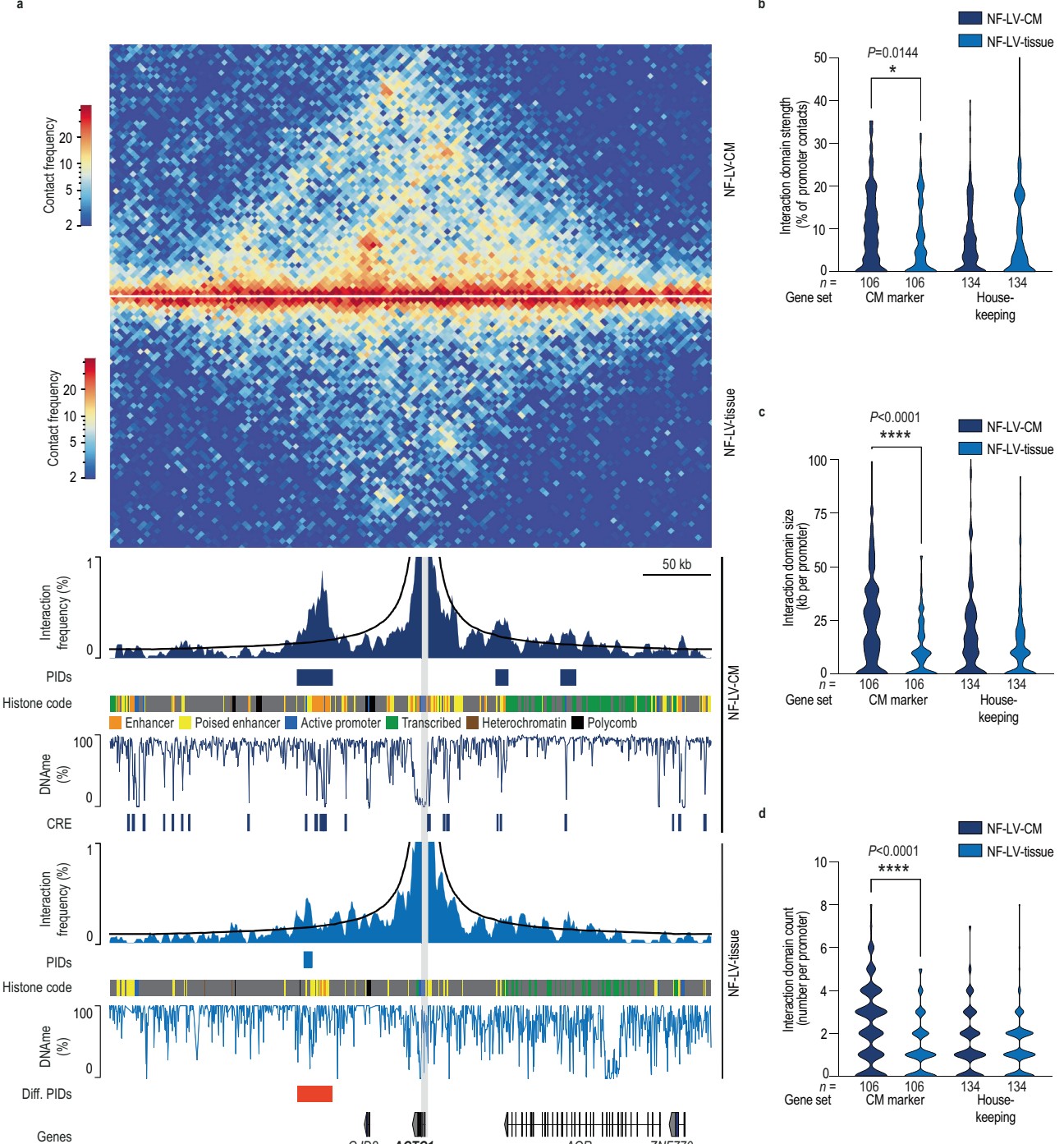

**Fig. 2 | Comparison of human cardiac tissue nuclei and cardiomyocyte Hi-C data. a** Original traces of chromatin interaction frequency heatmap at 5 kb resolution (Hi-C), virtual viewpoint analysis of *ACTC1* promoter interactions (1 kb resolution), mCpG (DNA methylation sequencing), and histone code (ChromHMM). Low-methylated *cis*-regulatory regions (CREs), PIDs, and differential PIDs are annotated. Gray area highlights *ACTC1* promoter. The shown Hi-C data sets are generated from the same biological replicates (*n* = 3). Diff. PIDs, Differential promoter-interacting domains. Violin plots of interaction frequency strength (**b**), length (**c**), and count (**d**) of PIDs for CM marker genes and housekeeping genes. **P* < 0.05, *****P* < 0.0001. Kruskal-Wallis test (**b**–**d**). Indicated n-numbers represent the number of genes measured for each group.

to non-CMs. In contrast, VISTA enhancers overlapping adult CM PIDs were linked to cardiac gene programs (Fig. 3e), with 14% residing within chamber-specific PIDs. A visual inspection of VISTA mouse embryos confirmed that many chamber-specific PIDs harbor enhancers with chamber-specific reporter gene activity (Fig. 3d). These findings demonstrate that PIDs drive chamber-specific gene expression programs, including key cardiac transcription factors.

## Rewiring of the NPPA promoter

Hi-C viewpoint analysis revealed robust and dynamic interaction between the *NPPA* promoter and a ~20 kb stretch of clustered H3K27ac-marked enhancers (super-enhancer, Fig. 5a, yellow box). The interaction strength between the *NPPA* promoter and super-enhancer was significantly higher in NF-LA-CM compared to NF-LV-CM (Fig. 5a, b). Notably, CpG methylation levels at the super-enhancer and adjacent

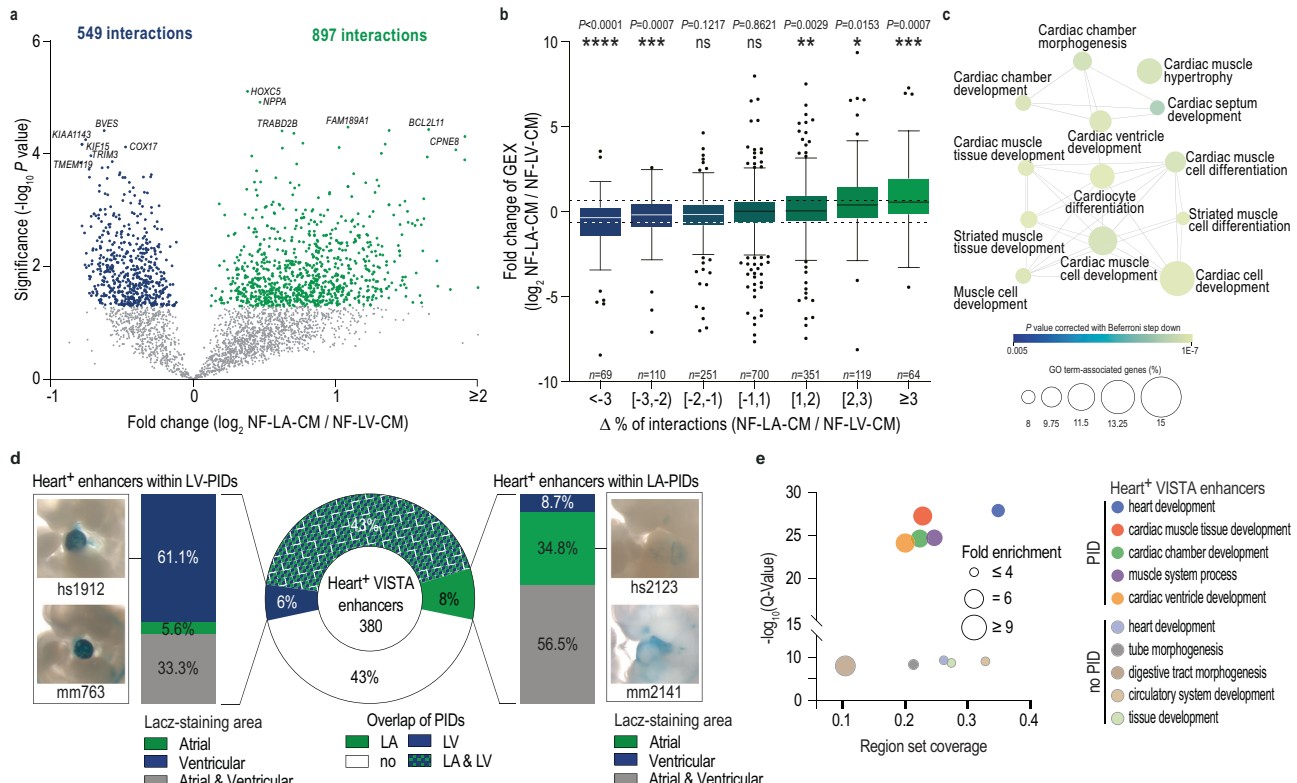

**Fig. 3 | Relationship between promoter interactions and gene expression of atrial and ventricular cardiomyocytes. a** Volcano plot illustrating changes in promoter-interacting domains (PIDs) between NF-LV-CM and NF-LA-CM. Differential PIDs were identified using a Chi-square test ($P < 0.05$), and a replicate-based post-hoc analysis was performed (two-sided unpaired t-test). Chromatin interactions with $P < 0.05$ are considered significant. Interactions significantly enriched in NF-LV-CM (blue points) and, NF-LA-CM (green points) are highlighted. Hi-C data is derived from 9 (NF-LV-CM) or 6 (NF-LA-CM) biological replicates. **b** Correlation between changes in promoter interactions and gene expression in NF-LV-CM and NF-LA-CM. Genes are grouped according to changes in differential interaction. Data presented as Box and whiskers (Tukey, line =median). *$P < 0.05$; **$P < 0.01$; ***$P < 0.001$; ****$P < 0.0001$; GEX gene expression. Two-sided one-sample Wilcoxon signed rank test. Annotated $n$ values represent the number of genes in each group. **c** Gene ontology analysis of differential promoter interactions ($P < 0.01$) between NF-LV-CM and NF-LA-CM. Significance threshold for calculating terms set at $P < 0.01$, Two-sided hypergeometric test with Bonferroni step-down correction. **d** Overlap of heart activity-positive (Heart[+]) VISTA enhancers with NF-LV-CM and NF-LA-CM PIDs (pie chart). Flanking plots visualize the localization of the reporter gene signal. The left and right bars summarize the localization for Heart[+] enhancers overlapping with NF-LA-CM PID or NF-LV-CM PIDs, respectively. Selected images of mouse embryos obtained from the VISTA database display chamber-specific reporter gene activity. **e** GREAT analysis of Heart[+] enhancers overlapping (saturated colors) or not overlapping (pale colors) PIDs. Fold enrichment and the $Q$ value of the enriched biological processes are shown.

regulatory elements showed no significant differences, suggesting that interaction strength, rather than CpG methylation, underlies chamber-specific *NPPA* expression.

Because *NPPA* upregulation is a hallmark of heart disease and is associated with epigenetic changes[11,16], we assessed chromatin rewiring at the *NPPA* locus in F-LV-CM. We identified PIDs in CM from failing hearts (F-LV-CM; Supplementary Data 6) and performed a replicate-based comparison to PIDs detected in NF-LV-CM. The *NPPA* promoter showed higher interaction frequencies with the super-enhancer region in F-LV-CM than in NF-LV-CM (Fig. 5a, c), mirroring the profile observed in NF-LA-CM. The increase in super-enhancer interactions was accompanied by enhanced *NPPA* expression and increased H3K27ac marking at the *NPPA* promoter, consistent with the transcriptional activation in failing CM. However, the histone modifications and CpG methylation at interacting CREs remained unchanged, suggesting that promoter-super enhancer rewiring underlies *NPPA* induction in failing CMs (Fig. 5a).

To further dissect the structure-function relationship of the *NPPA* super-enhancer regulatory landscape, we performed a tiled viewpoint analysis across the super-enhancer region (Fig. 5d). In F-LV-CM, the *NPPA* promoter primarily interacted with the proximal part of the super-enhancer, whereas NF-LA-CM showed the strongest interaction with its distal part (Fig. 5d), suggesting that different enhancer

segments mediate chamber-specific and disease-associated gene regulation.

We extended our analysis to the neighboring *NPPB* promoter, previously reported to interact with this super-enhancer, too ref. 44. In contrast to *NPPA*, the *NPPB* promoter specifically interacts with the central part of the super-enhancer in NF-LA-CM (Fig. 5e). Functional interrogation of this segment using AAV-CRISPRi (AAVi) in hiPSC-aCMs revealed that silencing the central super-enhancer segment reduced *NPPB* expression by 82% and *NPPA* expression by 31% (Fig. 5f, g).

These results indicate that distinct segments of the *NPPA*/*NPPB* super-enhancer orchestrate specific spatial and transcriptional control of gene expression in atrial and ventricular CMs, supporting a model in which distinct super-enhancer domains enable fine-tuned gene regulation across physiological and disease states.

**Promoter rewiring in failing CM**

Next, we conducted a genome-wide analysis of promoter rewiring in failing CMs and detected 4245 differential PIDs (Fig. 6a). Concordant changes in promoter interactions and gene expression (delta > 1.5 fold) were only detected for eight genes (*ATOH8, NPPA, GNG8, NRG1, FMOD, DACT3, GMPR,* and *SORT1*, Fig. 6a, b), including *NPPA* as described

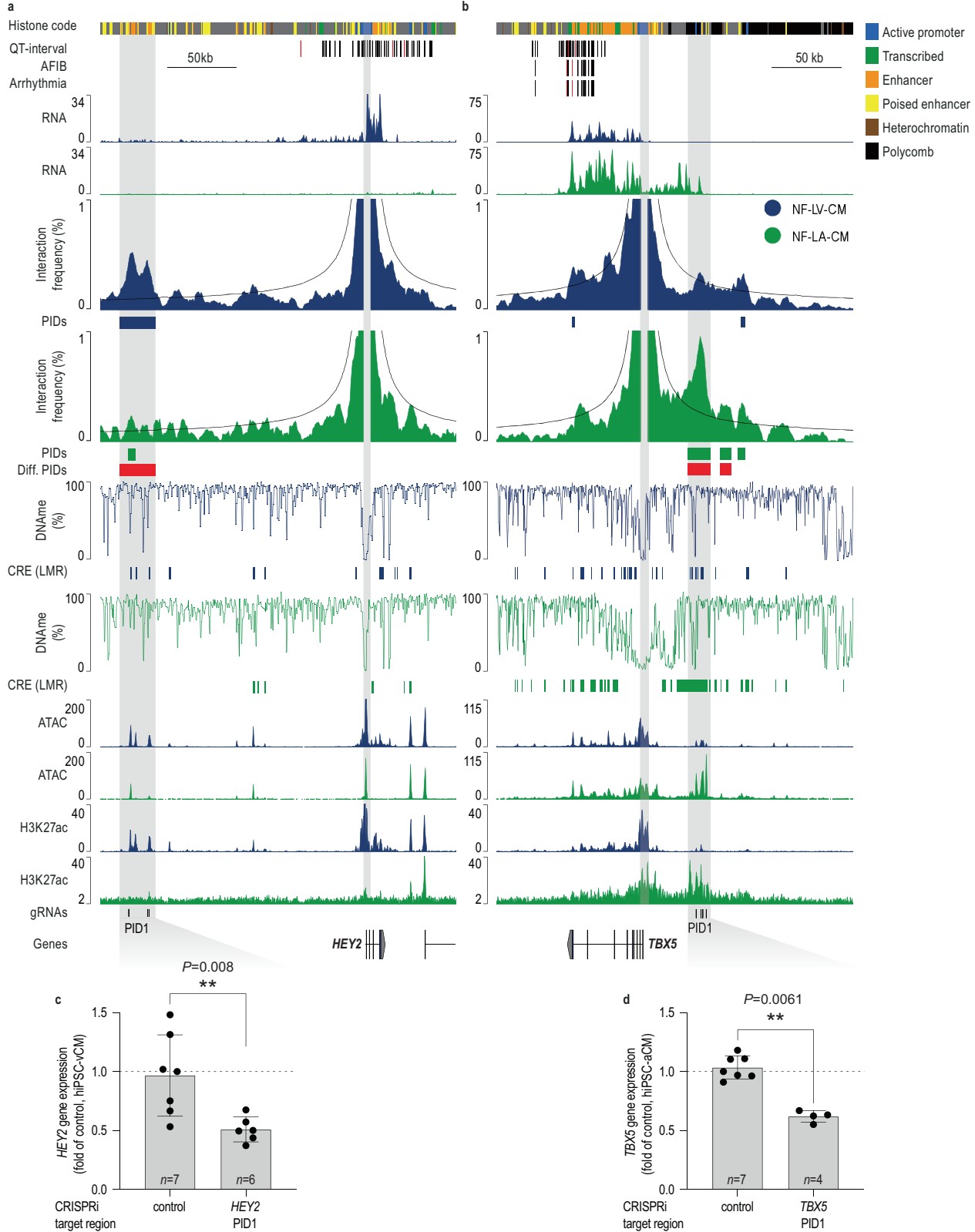

**Fig. 4 | Chamber-specific promoter interactions of *HEY2* and *TBX5* in atrial and ventricular cardiomyocytes.** Virtual viewpoint analysis of the *HEY2* (**a**) and *TBX5* promoters (**b**) at 1 kb resolution. Gene expression (RNA-seq), CpG methylation, chromatin accessibility (ATAC-seq), and histone code are shown. NF-LV-CM and NF-LA-CM tracks are shown in blue and green, respectively. Gray areas highlight the *HEY2* and *TBX5* promoters and the H3K27ac-marked enhancer. Hi-C data were derived from 9 (NF-LV-CM) or 6 (NF-LA-CM) biological replicates. sgRNA target regions for CRISPRi are annotated. AAV-mediated CRISPRi silencing of *HEY2* PID1 in hiPSC-vCM (**c**) and *TBX5* PID1 in hiPSC-aCM (**d**). Gene expression was measured by RT-qPCR. *n* indicated on the plot represents independent biological replicates. **P < 0.001; Two-tailed Mann-Whitney test. Data presented as mean ± SD.

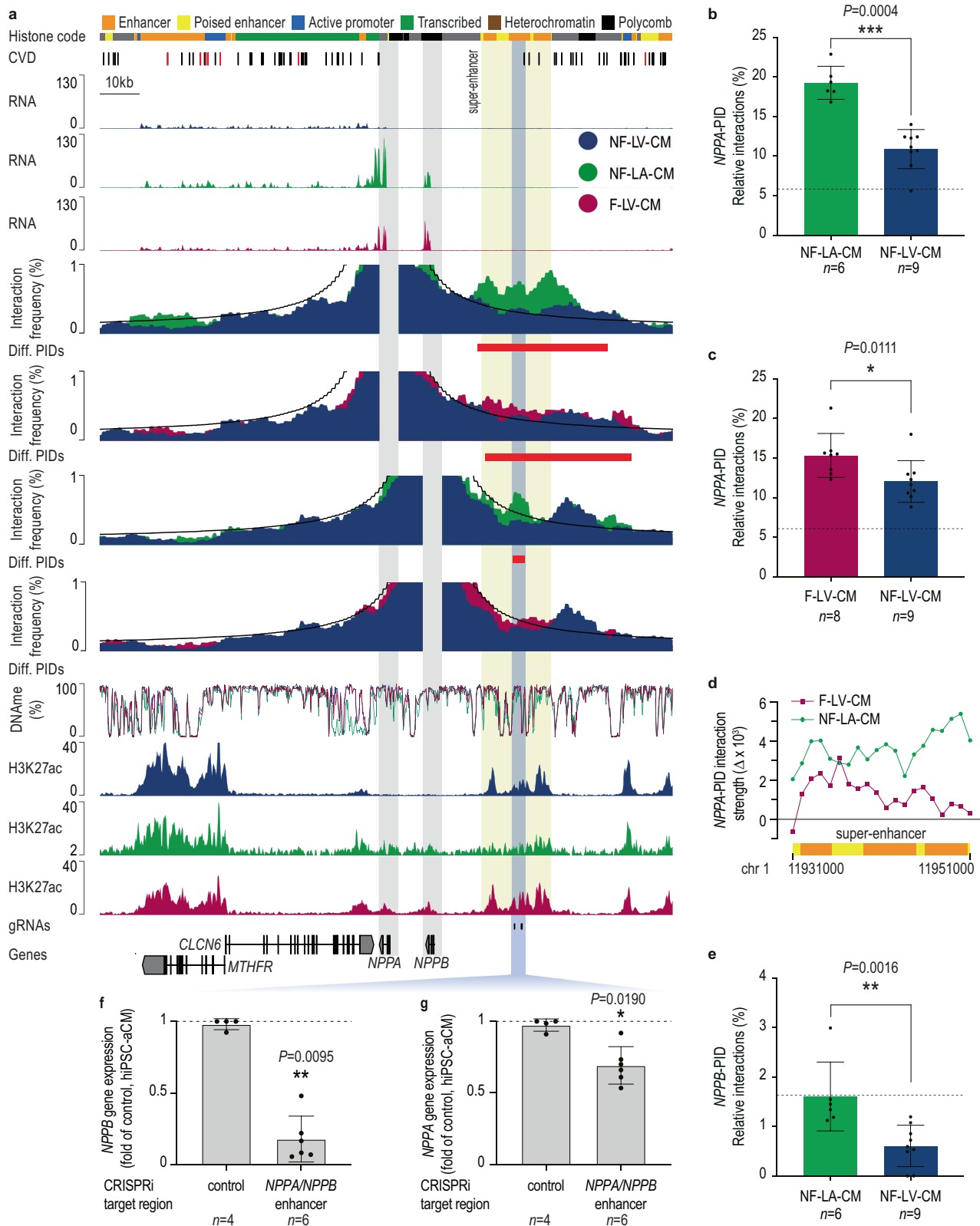

above (Fig. 5). A significant fraction of CREs losing promoter interaction strength in F-LV-CM contained a CTCF motif (Supplementary Fig. 19a). Integration of CTCF ChIP-seq data confirmed CTCF occupancy at these sites (Supplementary Fig. 19b). While CTCF is known to be implicated in TAD formation, TAD structure was preserved in failing CMs (Supplementary Fig. 11e, f). Differential PID interactions were constrained within TAD boundaries (Supplementary Fig. 19c), and only

2.9% of CTCF-marked differential PI-CREs were located at TAD boundaries.

We next asked whether chromatin interaction data link CRE activity to gene expression changes in failing CMs. Differential H3K27ac marking analysis identified two groups of CREs: group I (1848 CREs) with decreased H3K27ac and group II (1625 CREs) with increased H3K27ac levels (1626 CREs) in F-LV-CM (Fig. 6c). Linking these dynamic

**Fig. 5 | Rewiring and functional analysis of the *NPPA* and *NPPB* super-enhancer. a** Virtual viewpoint analysis of *NPPA* and *NPPB* promoter interactions (1 kb resolution), gene expression (RNA-seq), mCpG methylation, H3K27ac (ChIP-seq), and histone code (ChromHMM) are shown. NF-LV-CM, NF-LA-CM, and F-LV-CM tracks are shown in blue, green, and red, respectively. Gray areas highlight the *NPPA* and *NPPB* promoters. The H3K27ac-marked super-enhancer (yellow) and the CRISPRi-silenced region (blue) are highlighted. sgRNA target regions for CRISPRi are annotated. Hi-C data were derived from 9 (NF-LV-CM), 6 (NF-LA-CM), and 8 (F-LV-CM) biological replicates. **b, c** Comparison of relative *NPPA* promoter interactions with the differential PIDs overlapping the super-enhancer in NF-LV-CM, NF-LA-CM, and F-LV-CM. Data are presented as mean ± SEM. Two-tailed Mann-Whitney test. Dashed lines correspond to the background interactions. **d** A tiled viewpoint

analysis (1 kb tiles, step-size 1 kb) was performed. The relative *NPPA* promoter interactions are plotted. The ChromHMM annotation is visualized for the super-enhancer, and the color code is given in (**a**). **e** Comparison of relative *NPPB* promoter interactions with the differential PIDs overlapping the super-enhancer in NF-LV-CM and NF-LA-CM. Data are presented as mean ± SD. Two-tailed Mann-Whitney test. The dashed line corresponds to the background interactions. **f, g** AAV-mediated CRISPRi silencing of the central segment of *NPPA* and *NPPB* super-enhancer in hiPSC-aCM. Gene expression was measured by RT-qPCR. Two-tailed Mann-Whitney test. Data are presented as mean ± SD. **b–g** *n*-numbers indicated on the plots represent the number of biological replicates; *$P < 0.05$; **$P < 0.01$; ***$P < 0.001$.

---

CREs to differentially expressed genes via PIDs, revealed that CREs gaining H3K27ac (group II) were associated with upregulated genes, whereas those losing H3K27ac (group I) were associated with down-regulated genes in F-LV-CM (Fig. 6d). One representative example is *HEY2* (group I). The distal CRE stably interacting with the *HEY2* promoter in NF-LV-CM and F-LV-CM exhibited reduced H3K27ac signal and gene expression in F-LV-CM (Fig. 6e). Functional silencing of this CRE using CRISPRi in hiPSC-vCM recapitulated the downregulation observed in disease, confirming its regulatory role (Fig. 4d).

These results demonstrate that pathological gene expression changes rely primarily on preformed PIDs, and that pathological gene expression is linked to dynamic enhancer activity.

## Regulatory mechanisms of cardiac disease-associated non-coding elements

Several studies have established that cardiac CREs are enriched for cardiac disease-associated genetic variants[9,11,17,19,45–47]. However, as in most cells of our body[48], the target promoters of disease-associated CREs have not been systematically studied in a chamber- and CM-specific context. To address this, we analyzed the enrichment of disease-associated variants listed in the NHGRI-EBI GWAS catalog[49] within promoter-interacting (PI-CREs) versus non-interacting CREs (PN-CREs, Supplementary Fig. 13a) as visualized in Fig. 7a.

Enrichment analysis for different disease traits revealed that only cardiovascular, heart, and respiratory disease-associated variants were significantly enriched in PI-CREs of NF-LV-CM compared to random variants (Fig. 7b). Further stratification of heart disease entities showed a stronger enrichment of cardiomyopathy-, QT interval-, and heart failure-associated variants in PI-CREs than PN-CREs of NF-LV-CM (Fig. 7c). The strongest genetic association of PI-CREs detected in NF-LV-CM was found for diseases that primarily affect the ventricles (Fig. 7c). In contrast, atrial fibrillation (AFIB)-associated variants displayed minimal enrichment in these NF-LV-CM PI-CREs. We then asked whether genetic variants associated with AFIB are enriched in PI-CREs of NF-LA-CM. Indeed, AFIB-associated variants were enriched in PI-CREs of NF-LA-CM ($P = 5.9 \times 10^{-22}$) as compared to NF-LV-CM (Fig. 7d, e). Similar trends were noted for heart disease and arrhythmia GWAS categories, largely reflecting the prevalence of AFIB-associated variants in them. In contrast, PI-CREs of NF-LV-CM carry more risk alleles for QT interval, cardiomyopathy, and heart failure ($P = 1.3 \times 10^{-46}$, $2.2 \times 10^{-10}$, $1.6 \times 10^{-5}$, respectively) as compared to PI-CREs detected in NF-LA-CM. Next, we confirmed the benefit of our chromatin interaction maps for disease variant interpretation using two recently published dilated cardiomyopathy (DCM)[50] and QT[51] GWAS studies. Consistent with our analysis of NHGRI-EBI GWAS catalog data[49], NF-LV-PIDs were significantly enriched for DCM- and QT interval-associated variants (Fig. 7c-e). For this analysis, we selected variants meeting the criteria of GWAS ($P < 5 \times 10^{-8}$). To evaluate the potential of chamber-specific PI-CRE maps in annotating sub-threshold risk variants, we expanded this analysis to sub-threshold variants reported by Nauffal[51] ($P < 0.1$) and performed an enrichment analysis (Fig. 7f). Strikingly, NF-LV-CM PI-CREs (Fig. 7f) were already strongly enriched for QT variants

at significance levels of $P < 10^{-5}$, while NF-LA-CM showed minimal enrichment independent of the chosen *P* value cut-off (Fig. 7f). CRE variants meeting GWAS or sub-threshold criteria were linked to genes of comparable gene ontology classes, including cardiac muscle cell membrane repolarization (GO: 0099622). 86% of QT-associated CRE-promoter interactions spanned at least one other gene promoter, indicating that the nearest gene was often not the regulatory target. This underscores the importance of integrating chromatin interaction maps for ascribing biological functions to non-coding genetic variants (Supplementary Fig. 13h). The sub-threshold cut-off criteria identified 25% more genetic variants of PI-CREs as compared to the GWAS criteria for this ontology term in NF-LV-CM (Fig. 7g). These findings demonstrate that integrating chamber-specific PI-CRE data enhance variant-to-gene mapping and supports the discovery of additional disease-associated regulatory loci.

## Functional validation of disease-associated enhancers

To show the functional relevance of disease-associated PI-CREs we took advantage of published cardiac "expression quantitative trait loci" (eQTL) data[52]. Chamber-specific PIDs linked genetic variation to gene expression changes in human hearts, supporting the functional relevance of PI-CREs. This included genetic variants spatially interacting with AFIB and long QT syndrome-associated genes such as *KCNK3* and *NOS1AP*[53,54] (Fig. 7h–j). We next aimed to test the functional relevance of individual disease-associated PID-CREs using CRISPRi in hiPSC-CM. We selected several CREs genetically linked to QT duration due to the strong enrichment of this trait in PI-CREs of NF-LV-CM (Fig. 7c–f). CRISPRi-mediated silencing of two QT-linked CREs reduced gene expression of their respective target genes *KCNH2* and *SCN5A* in hiPSC-vCM (Supplementary Fig. 14a and Supplementary Fig. 15), consistent with previous reports[19,37].

We next aimed to decode the regulatory landscape of *KCNJ2*, since it harbors several uncharacterized CREs carrying QT variants, including CREs overlapping with variants meeting genome-wide significance ($P < 5 \times 10^{-8}$) and sub-threshold ($P < 10^{-5}$) variants. We restricted this analysis to active CREs marked by H3K27ac in LV-CM (Fig. 8a, Supplementary Fig. 20). *KCNJ2* encodes the inward-rectifier potassium channel Kir2.1, a key contributor to the $I_{K1}$ current. We first tested whether our matured hiPSC-vCM expressed functional levels of *KCNJ2*. *KCNJ2* transcript levels were higher in hiPSC-vCM than in hiPSC-aCM but remained lower than in native LV tissue (Supplementary Fig. 21a), consistent with previous reports[55,56]. Automated high-throughput patch-clamp experiments detected Kir2.1 currents, reflecting functional KCNJ2 channels, in 72.5% of hiPSC-vCM (Supplementary Fig. 21b–d). This aligns with our previous observations[57] and confirms the suitability of hiPSC-vCM for functional studies. Using CRISPRi, we systematically silenced six *KCNJ2* interacting domains (regions 3–8), one non-interacting (region 2), and the *KCNJ2* promoter (region 1, positive control) in hiPSC-vCM. Silencing of the *KCNJ2* promoter led to near-complete (98.4%) loss of gene expression (Fig. 8b), while silencing of the non-interacting region 2 or the interacting region 8 did not significantly

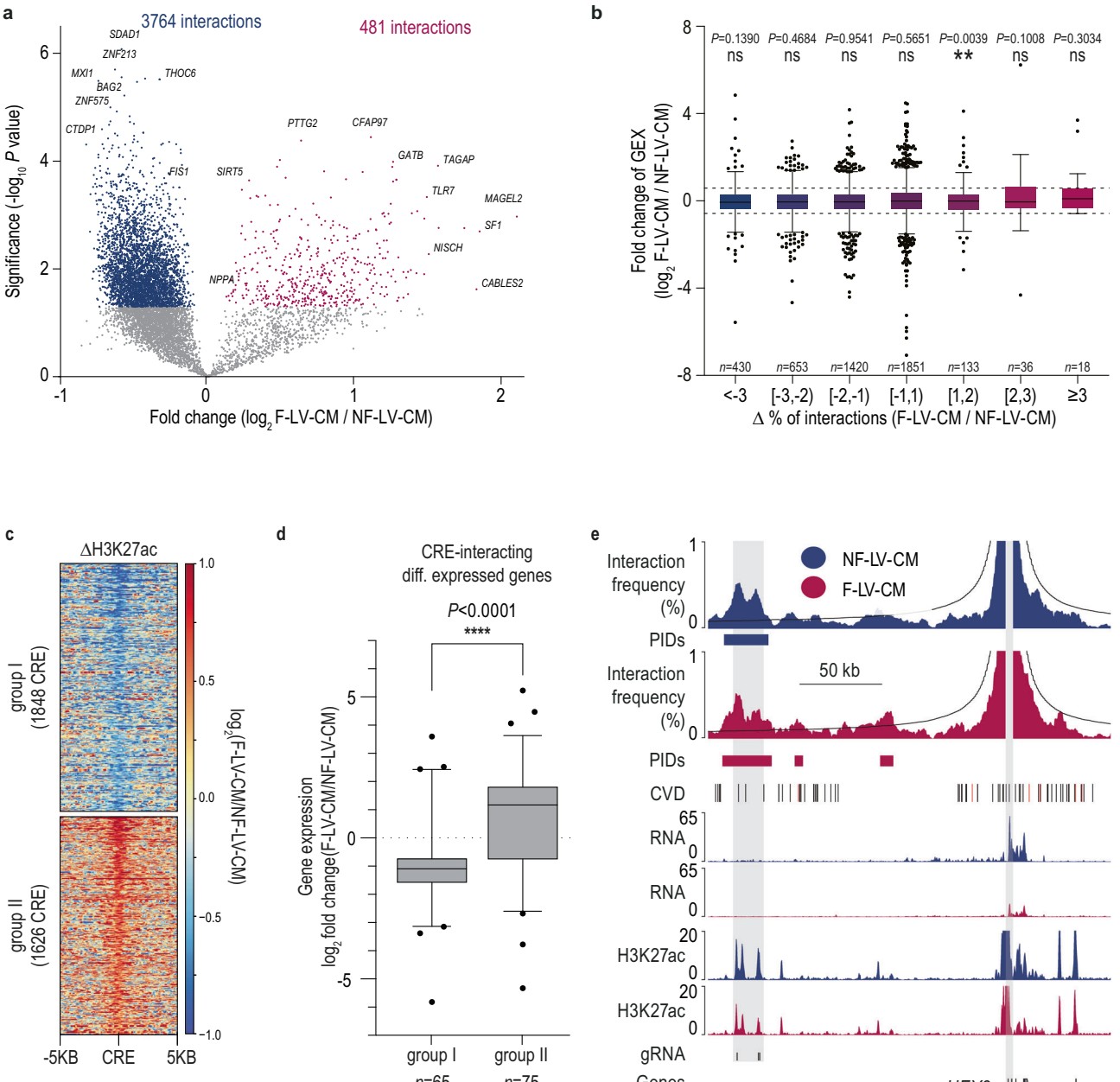

**Fig. 6 | Chromatin interaction dynamics and enhancer remodeling in failing cardiomyocytes. a** Volcano plot illustrating changes in promoter-interacting domains (PIDs in NF-LV-CM vs F-LV-CM). Differential PIDs were identified using a Chi-square test ($P < 0.05$), and a replicate-based post-hoc analysis was performed (two-tailed unpaired t-test). Chromatin interactions with $P < 0.05$ are considered significant. Interactions significantly enriched in NF-LV-CM (blue points) and F-LV-CM (red points) are highlighted. Hi-C data derived from 9 (NF-LV-CM) or 8 (F-LV-CM) biological replicates. **b** Bar plot showing the relationship between changes in promoter interactions and gene expression in NF-LV-CM versus F-LV-CM. Genes are categorized based on promoter interaction dynamics. Data presented as Box and whiskers (Tukey, line = median). Dotted lines correspond to a 1.5-fold change in GEX. **$P < 0.01$; Two-tailed one-sample Wilcoxon signed rank test. GEX, gene expression. $n$-numbers indicated on the plots represent the number of genes per group. **c** Analysis of PI-CREs with differential H3K27ac-marking between NF-LV-CM and F-LV-CM. Heatmaps are grouped according to decreased (group I) and increased (Group II) H3K27ac-marking in F-LV-CM as compared to NF-LV-CM. **d** Linking differential H3K27ac-marking of for group I and II PI-CREs to differential gene expression. Data presented as box and whiskers (box = 25th–75th percentile, line = median, and whiskers = 5th–95th percentile). ****$P < 0.0001$; Two-tailed Mann-Whitney test. $n$-numbers indicated on the plots represent the number of PI-CRE. **e** Virtual viewpoint analysis of *HEY2* promoter interactions (1 kb resolution), gene expression (RNA-seq), H3K27ac (ChIP-seq), and histone code (ChromHMM) are shown. NF-LV-CM and F-LV-CM tracks are shown in blue and red, respectively. Gray areas highlight the *HEY2* promoter and the H3K27ac-marked super-enhancer. gRNA target regions for CRISPRi (see Fig. 4c) are annotated.

impact *KCNJ2* expression. The silencing of interacting regions 3, 4, and 6 resulted in 39.7%, 65.4%, and 50.5% reduction of *KCNJ2* gene expression, respectively (Fig. 8b), confirming their enhancer activity. Two independent genetic deletions of region 4 validated these findings (Supplementary Fig. 22). Combinatorial silencing of adjacent regions 3 and 4 (Fig. 8c) did not find a significant additive

effect. Chromatin accessibility analysis (ATAC-seq) for their individual and combinatorial silencing (Fig. 8d) demonstrated the specificity and efficacy of CRISPRi-mediated silencing. Chromatin interaction analysis demonstrated that CRISPRi-mediated silencing did not affect the interaction strength of regions 3 and 4 with the *KCNJ2* promoter (Supplementary Fig. 14f). Given that regions 3 and

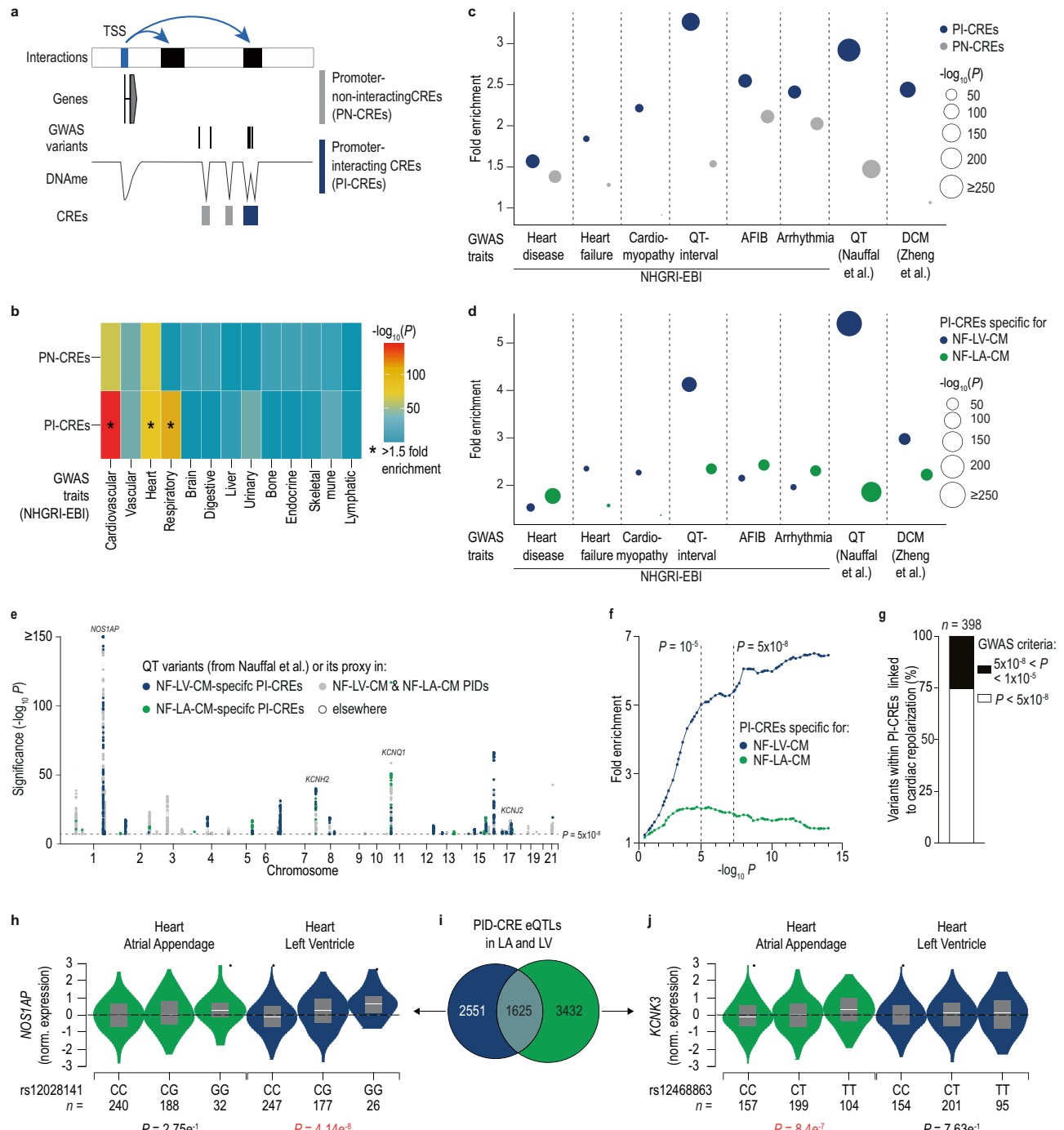

**Fig. 7 | Linking non-coding risk factors to target promoters in atrial and ventricular CM. a** Strategy to categorize disease-associated variants according to overlap with promoter-interacting CREs (PI-CREs) and promoter-non-interacting CREs (PN-CREs). **b** Heatmap displaying the significance of the enrichment calculated for genome-wide association study (GWAS) variants compared to random variants for NF-LV-CM PI-CREs and PN-CREs across various organ disease traits from the NHGRI-EBI GWAS catalog. Stars mark enrichments exceeding 1.5-fold. One-side Enrichment $P$ value with Bonferroni correction is shown. **c** Enrichment analysis comparing the overlap of disease-associated variants with PI-CREs and PN-CREs in NF-LV-CM for different heart disease traits from the NHGRI-EBI GWAS catalog and two published studies[50,51]. One-side Enrichment $P$ value with Bonferroni correction is displayed. **d** Enrichment analysis as described in **c** for ventricular-specific PI-CREs and atrial-specific PI-CREs. **e** Manhattan plot showing the overlap between CREs and QT-interval-associated variants. Variants present in atrial and ventricular PIDs

are highlighted in gray. Variants in ventricular-specific PI-CREs or atrial-specific PI-CREs are highlighted in blue or green, respectively. Shown are variants with a genome-wide association cut-off ($P < 5 \times 10^{-8}$). **f** Enrichment analysis of QT variants was performed for chamber-specific PI-CREs. The curves display enrichment for different $P$ value cut-offs. **g** Number of genetic variants spatially interacting with cardiac muscle cell repolarization-associated gene promoters (GO: 0099622) fulfilling reported sub-threshold or GWAS $P$ value criteria[51] in CREs of NF-LV-CM. **h–j** Non-coding genetic variants significantly correlated with gene expression (eQTL) detected in ventricular tissue and atrial appendage[52] overlap with PI-CREs of NF-LV-CM and NF-LA-CM, respectively. Venn diagram (**i**) shows the overlap of eQTLs and PI-CRE pairs. Violin plots display representative eQTL and PI-CRE pairs with chamber-specific effects on *NOS1AP* (**h**) and *KCNK3* (**j**) expression. The overlaid box plot indicates median and 25th and 75th percentiles. The indicated n-number represents the number of biological replicates per group.

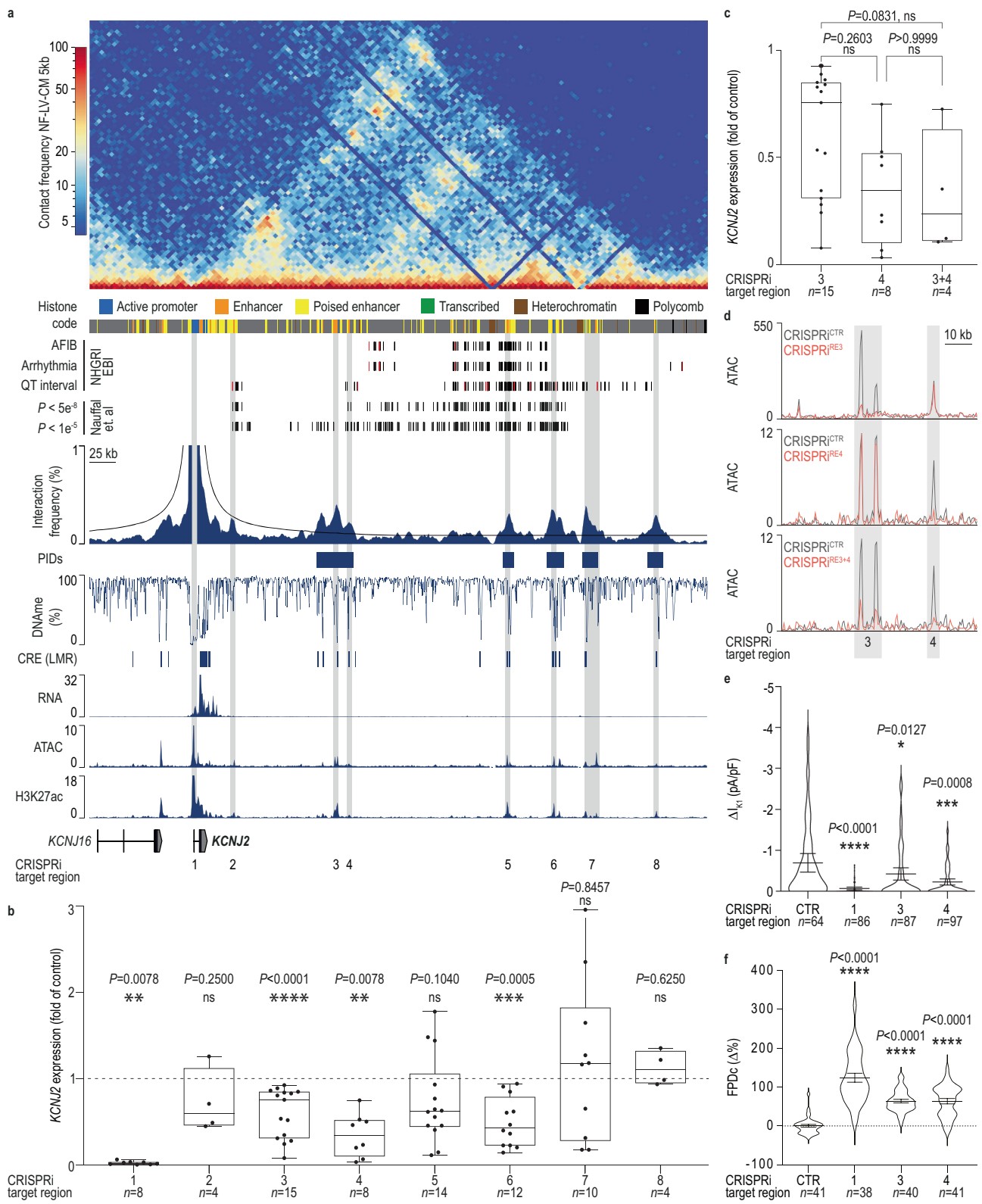

4 are enriched in disease risk variants and critical for *KCNJ2* transcription, we hypothesized that their modulation, by affecting the level of functional KCNJ2, would also impact the inward-rectifier $I_{K1}$ currents. Silencing of the *KCNJ2* promoter resulted in a 90% reduction of $I_{K1}$ currents, whereas silencing of regions 3 and 4 reduced $I_{K1}$ currents by 39% and 67%, respectively, closely mirroring the transcriptional effects (Fig. 8e, Supplementary Fig. 21b–f). Multi-electrode arrays (MEA) recordings of field potentials further

demonstrated that silencing these regions in hiPSC-vCMs delayed repolarization time, mimicking QT interval prolongation (Fig. 8f).

Collectively, the dissection of the *KCNJ2* locus identified key enhancer elements modulating its expression levels and the repolarization kinetics in CMs, providing mechanistic insights into the genetic basis of QT syndrome. Notably, the functionally relevant region 3 does not overlap with QT duration-associated GWAS variants but instead harbors a sub-threshold variant (Fig. 8a).

**Fig. 8 | Functional relevance and promoter interactions of disease-associated non-coding variants in the *KCNJ2* locus. a** Heatmap of chromatin contact frequency (5 kb resolution, *n* = 9) and GWAS variants from selected NHGRI-Ebi GWAS catalog entities[49] (lead-SNPs = red, proxy-SNPs = black). The GWAS *P* values from Nauffal et al.[51] were used to extract QT interval-linked variants fulfilling stringent ($5 \times 10^{-8}$) and sub-threshold ($10^{-5}$) GWAS *P* value cut-offs. Virtual viewpoint analysis for the *KCNJ2* promoter (1 kb resolution) and chromatin states are shown. Original traces for gene expression (RNA-seq), chromatin accessibility (ATAC-seq), and H3K27ac (ChIP-seq) of NF-LV-CM. Gray areas highlight CRISPRi silenced regions. **b** CRISPRi-mediated silencing of *KCNJ2* promoter-interacting elements (regions 3–8), non-interacting regulatory element (region 2), and the *KCNJ2* promoter (region 1) using lentiviral transduction of hiPSC-vCM. Gene expression was quantified by RT-qPCR. Data are shown as box and whiskers (box = 25th–75th percentile, line = median, whiskers = minimum to maximum). *n*-numbers indicated on the plot represent independent biological replicates. ***P* < 0.001; Two-tailed one-sample Wilcoxon test. **c** *KCNJ2* gene expression analysis following combinatorial CRISPRi-mediated targeting of two regulatory elements (region 3 and 4). Gene expression of lentiviral-transduced hiPSC-vCM was measured by RT-qPCR. Values for single-region CRISPRi targeting regions 3 and 4 were replotted from Fig. 8b. Data are

shown as box and whiskers (box = 25th–75th percentile, line=median, whiskers = minimum to maximum). *n*-numbers indicated represents independent biological replicates. ns, *P* ≥ 0.05; Kruskal-Wallis test with Dunn's adjustment for multiple comparisons. **d** Chromatin accessibility changes assessed by ATAC-seq after CRISPRi silencing of region 3 and 4 individually or in combination. Control tracks (black) represent corresponding non-targeting CRISPRi conditions. Region 3 was silenced using AAV-CRISPRi, while region 4 and the combined regions 3 and 4 were silenced using lenti-CRISPRi. Shown are merged data from *n* = 3–4 biological replicates. **e** Measurement of inward rectifier potassium current ($I_{K1}$) in hiPSC-vCM after AAV-mediated CRISPRi silencing of *KCNJ2* region 1, 3, and 4 using automated patch-clamp. Data are normalized to the cell capacitance (pA/pF) and shown as mean ± 95% CI. *n*-numbers indicated on the plot represent individual cells. Kruskal-Wallis test with Dunn's adjustment for multiple comparisons after outlier exclusion (ROUT, Q = 1%), **P* < 0.05; ***P* < 0.001 as compared to control. **f** Field potential durations (FPD) were measured in hiPSC-vCM after AAV-mediated CRISPRi silencing of *KCNJ2* region 1, 3, and 4 using multi-electrode arrays and corrected using the Fridericia correction formula (FPDc). *n*-numbers indicated represent independent biological replicates. (mean ± SEM). Kruskal-Wallis test with Dunn's adjustment for multiple comparisons ***P* < 0.001 as compared to control.

Finally, we note that the respective promoter interactions are barely detectable in LV-tissue Hi-C data (Supplementary Fig. 4), emphasizing the benefit of cell-type-specific epigenome and chromatin-interaction profiling, combined with functional modulation, to decode the disease relevance of non-coding regulatory elements.

## Discussion

Distal regulatory elements are essential for transcriptional control in mammals. These regulatory elements are extensively conserved between mammalian species. However, only a fraction of the more than a million predicted regulatory elements in the human genome is established in a specific cell type or state[3,28]. In the heart, previous DNA methylation-guided annotation of *cis*-regulatory elements across major murine cardiac cell types, including CMs, fibroblasts, and endothelial cells, found that more than 30–50% are cell-type-specific[18]. Although CMs account for most of the cardiac muscle and RNA mass, they represent only about one-third of all cardiac cells[11,13,58]. The present work highlights the value of CM isolation for accurate annotation and characterization of their specific regulatory elements and target genes, consistent with insights from recent single-cell epigenetic studies [59].

Several studies, including this one, charted the regulatory landscape of human CMs[11,17], identifying over 100,000 CREs. These elements are enriched for disease-associated variants[11]. Cell-type-specificity of CREs and their dynamic spatial interactions with target genes may explain the penetrance and phenotype of genetic risk factors. Chromatin capture data of in vitro differentiated CMs and heart tissue[9,14,22,45,60,61] have shown enrichment of variants associated with heart disease traits in promoter-interacting regulatory elements. Integrative analysis of epigenetic marks and chromatin accessibility has allowed prioritizing candidate genetic regulatory variants, some of which have been confirmed using enhancer deletions in ES- and iPSC-derived CMs[9,17,19,22,62]. However, they do not allow CM-specific and chamber-specific association analysis of mature human CMs. Furthermore, although iPSC-derived CMs are a valuable model of cardiac development and homeostasis, they do not fully recapitulate the transcriptional and functional properties of adult and mature CMs.

Here, we provide a high-resolution genome-wide map of chromatin interactions in human adult CMs. A comparative analysis of the atrial and ventricular CM regulatory landscapes showed chamber-specific enrichment of variants associated with heart disease traits in promoter-interacting domains. For example, atrial fibrillation-associated variants are more prevalent in promoter-interacting CREs of atrial than ventricular CMs. Notably, integrating specific Hi-C data

was essential for uncovering these links, since earlier studies analyzing regulatory elements in atrial and ventricular CMs did not report such a chamber-specific enrichment[19].

A previous study demonstrated that integrating epigenetic annotations with cardiac GWAS data allows the identification of additional disease-associated variants that do not meet the genome-wide significance criteria[44]. Our findings confirm and allow us to extend this approach: by incorporating CM-specific genome-wide promoter-interaction maps and epigenetic data for left atria and ventricles, we can show that QT duration-associated variants from recent studies[49,51] are specifically enriched in ventricular promoter-interacting CREs, explaining their effect on QT duration as one of the main risk factors of ventricular arrhythmias, including sudden cardiac arrest. A large fraction of sub-threshold genetic variants is missed in the original GWAS when not accounting for the CM genome's regulatory context. Our data further link genetic variation to CREs in CMs and to interacting target genes. This knowledge is essential for understanding the underlying mechanisms of genetic risk factors. Thus, integrating cell-type-specific epigenetic and chromatin interaction data as a weighting factor into the analysis of GWAS data will uncover variants that modulate the function of regulatory elements, their impact on transcriptional control, and the onset of disease. Due to the multiple comorbidities and lifestyle-associated risk factors that mutually impact the onset and progression of heart disease entities, including the interrelationship between arrhythmia and heart failure[63,64], genetic association studies are challenging. The integration of epigenetic and chromatin interaction data links genetic variants to their distal target genes, thereby improving the interpretation of GWAS data and identification of disease mechanisms. We prioritized genetic variants associated with different disease entities using our data. Our representative integrative analysis of recent QT interval GWAS data[51] assigns biological effects even to sub-threshold variants. This prediction was functionally confirmed for a variety of genetically linked CREs, including ones interacting with the *KCNJ2*, *SCN5A*, and *KCNH2*[19] promoters. At the *SCN5A* locus, we identified a regulatory region within a super-enhancer whose mouse ortholog had previously been shown to control *Scn5a*[37], revealing the functional conservation of this domain across species. At the *KCNJ2* locus, functional dissection of the regulatory landscape identified three CREs that significantly modulate *KCNJ2* expression. For two of these, we demonstrated the impact on electrophysiological properties, providing insights into the putative mechanism of action of CREs genetically linked to the QT interval[51]. In the case of *KCNH2*, a CRE affected its expression and function as reported previously[19].

Previous studies used CRISPR-mediated deletion of CREs for functional studies[9,17,19]. Our data indicate that CRISPRi is an alternative approach, since CRISPRi precisely silenced individual regulatory elements. This aligns with a recent study showing that CRISPRi-mediated silencing establishes heterochromatin in a 5 kb range in CMs and that the effect does not spread to spatially interacting domains[34]. CRISPRi also did not affect the interaction of the silenced regulatory elements with the target promoter. A comparable observation was recently published for CRISPRa, where CRE activation did not affect 3D genome organization[65].

In contrast, it has been reported that targeting CTCF-bound TAD borders using a DNMT3A-based CRISPRi construct results in a loss of CTCF-mediated chromatin looping[66].

Our data further suggest that CTCF-mediated chromatin interactions are affected in terminal heart failure in CM. Garrido and coworkers[67] showed that reverse remodeling of heart failure is associated with elevated CTCF protein levels in the human heart. Several attempts to explore the role of CTCF and cohesin in maintaining chromatin structure and function in heart disease in mice resulted in contrasting observations[68,69]. The role of CTCF-associated rewiring in heart disease and its association with pathological gene expression thus remains unresolved from our point of view. Previous studies have also shown that the induction of *NPPA* in heart failure involves a loss of repressive polycomb marks and a gain in active chromatin marks of the *NPPA* promoter, including H3K27ac and H3K4me3. In contrast, the epigenetic landscape of the super-enhancer region remains stable[11,70]. Our results support that increased promoter interactions with distal regulatory elements and the establishment of active chromatin at the genic region of *NPPA* drive gene expression in diseased CMs. We detected a high contact frequency between this super-enhancer and both the *NPPA* and *NPPB* promoters in atrial CMs, too. Silencing of the central segment of the super-enhancer reduced expression of *NPPA* and *NPPB*, indicating its functional importance. The positive association of gene expression and CRE interaction frequency was not restricted to *NPPA* and *NPPB*. Altered promoter-CRE interactions were correlated with gene expression in atrial versus ventricular CM on a global scale, with target genes essentially related to heart development and contraction.

In summary, combining cell-type-specific chromatin interaction maps with CRISPRi-based functional assays is a robust approach for identifying and characterizing distal regulatory elements. The power of this strategy is underscored by the identification of *cis*-regulatory elements that control *KCNJ2* expression and its associated electrophysiological functions. These *cis*-regulatory elements harbor genetic variants associated with the onset of ventricular arrhythmias. In line with these observations, the genome-wide epigenetic and spatial interaction maps presented here will facilitate the interpretation of GWAS studies to uncover novel genetic variants driving heart disease and ascribe their biological functions.

## Limitations

Sample variability stemming from diverse genetic backgrounds and comorbidities is a common challenge in studies including human samples. We addressed this by including at least six biological replicates per group and generating epigenetic and transcriptomic data from matching biological samples. This study was restricted to male donor biopsies due to the unavailability of age-matched female control tissue. While the applied replicate-based differential chromatin interaction analysis increases statistical robustness, it can miss rewiring events occurring in a small subset of samples. Detecting these rare chromatin rewiring events would require increasing the sample size by several orders of magnitude. Alternatively, rewiring may be restricted to specific CM subpopulations. A recent preprint, reporting single-cell chromatin interaction data[71], does not report individual promoter-CRE pairs, likely due to the limited resolution per nucleus and or cell cluster, further highlighting the challenges in resolving such events.

We used hiPSC-CM to functionally validate candidate CREs. Although a comparative analysis revealed extensive similarities, hiPSC-CM remain immature in many respects[72]. This extends, to some degree, to the epigenetic layer as highlighted by a recent snATAC-seq study comparing fetal hearts with hiPSC-CM[22]. To exclude major biases, we confirmed that the regulatory landscape of perturbed loci is comparable between human adult and hiPSC-derived CM. Furthermore, CRISPRi perturbation only addresses the activity of the targeted CREs, at the tested time point – typically a differentiated CM state – and does not capture temporal dynamics during cardiomyocyte differentiation and maturation. Moreover, it does not directly test the actual consequence of individual disease-associated variants located within these elements.

Biases arising from variability in wet-lab protocols, bioinformatic processing pipelines, and overall data quality represent a broader limitation of epigenetic studies that integrate multiple data modalities. To minimize such confounders, we restricted our analysis to datasets generated using the same methodological approaches and re-analyzed all data from raw reads using unified workflows. Only datasets meeting rigorous and high-quality standards were included in the analysis.

## Methods

### Human cardiac biopsies

Cardiac biopsies from male hearts were used to isolate CM nuclei. These investigations were approved by the ethics committees of the Universities of Heidelberg, Munich, and Freiburg (Germany). Informed consent of human participants was obtained for surgical biopsies. Informed consent was waived by the ethical committee for samples from accidental deaths. Samples were not used if a participant or next of kin dissented from the scientific use. The study complies with the Declaration of Helsinki. All samples retrieved during interventions were immediately flash-frozen and stored at −40 to −80 °C. Tissue from accidental death was flash-frozen during the forensic autopsy. To comply with ethics committee approval requirements and protect patient data security, we removed data that could identify patients, including patient-specific sequence variation. Measurements of each biological replicate were taken from distinct samples listed in Supplementary Data 2. All tissue samples were fully used for experimental analysis.

### FANS of cardiomyocyte nuclei

All steps were performed at 4–8 °C to ensure nuclei integrity[13]. Frozen tissue (<100 mg) was thawed in 3 mL lysis buffer (5 mM CaCl2, 3 mM MgAc, 2 mM EDTA, 0.5 mM EGTA, 10 mM Tris-HCl, pH 8) and homogenized using a Miltenyi gentleMACS dissociator equipped with M tubes using the "protein_01" protocol. 3 mL of lysis buffer with 0.4 % Triton X-100 was added. The suspension was filtered through a 40 μm cell strainer (BD Bioscience), and the filter was washed with 2 mL lysis buffer. The suspension was centrifuged (1000 × *g*, 5 min, 4 °C), and the pellet was resuspended in 1 mL lysis buffer. The suspension was overlayed onto a 1 M sucrose cushion (3 mM MgAc, 10 mM Tris-HCl, pH 8) and centrifuged (1000 × *g*, 5 min, 4 °C). The pellet was resuspended in 500 μl staining buffer (PBS, 1 % BSA, 22.5 mg/mL glycine, and 0.1% Tween20) containing rabbit anti-PCM-1 antibody (1:1000, HPA023374, Sigma-Aldrich) and mouse anti-PLN antibody (1:1000, A010-14, Badrilla). After 30 min incubation at 4–8 °C, anti-rabbit secondary antibody conjugated to Alexa 568 (1:1000, A11011, Life Technologies) and anti-mouse secondary antibody conjugated to Alexa 488 (1:1000, A11029, Life Technologies) were added and incubated for an additional 30 min. After centrifugation (1000 × *g*, 5 min, 4 °C), the nuclei were resuspended in PBS buffer with Draq-7 (1:100, Cell Signalling Technology). CM nuclei

(PCM1+, PLN+, Draq-7+) were sorted using a SH800S cell sorter (Sony Biotechnology). FSC pulse width was used to exclude doublets from sorting, and PCM1- and PLN- double positive nuclei were selected. Sorted CM nuclei were processed immediately for downstream applications.

## Methyl-seq

Genomic DNA was extracted from the FANS-derived CM nuclei using AllPrep DNA/RNA Micro Kit (Qiagen, 80284). 200 ng of isolated DNA was sheared to ~300 bp using a Bioruptor Pico (Diagenode, 10 cycles, 30 s on, 90 s off, "low"). Sequencing libraries were subsequently constructed using the NEBNext Methyl-seq Kit (New England Biolabs, E7120), following the Protocol for Standard Insert Libraries.

## Hi-C

At least 25,000 sorted CM nuclei (up to 225,000 nuclei per technical replicate) or 20,000–50,000 hiPSC-vCMs were used for in situ Hi-C using the Arima Genome-wide HiC Kit and processed according to the manufacturer's documentation. DNA fragmentation was performed with a Bioruptor (Diagenode, 30 cycles, 30 s on, 90 s off, 'low') to obtain an average fragment size of 400 bp. The final Hi-C library was amplified using the KAPA-SYBR Fast DNA amplification reagents (Kapa Biosystems) supplemented with EvaGreen (Biotium) in technical replicates. The final sequencing libraries were constructed using the Swift Bioscience indexing kit according to instructions provided by Arima. Technical replicates were pooled and cleaned-up twice using 0,9x Ampure Beads XP (Beckman) to remove contaminating adapter and primer dimers.

## ChIP-seq

A minimum of 5000 sorted CM nuclei were used for ChIP-seq according to the manufacturer's protocol (Low Cell ChIP Kit, Active motif, 53086) using an antibody targeting H3K27ac (Active motif, 39133). Libraries were prepared using the NEBNext Ultra II DNA Library Prep Kit for Illumina (NEB, E7645) according to manufacturer's instructions. We monitored library amplification upon addition of EvaGreen using a qPCR cycler (Agilent). Amplification was stopped when the fluorescent signal reached the inflection point.

## hiPSC differentiation into atrial and ventricular cardiomyocytes

Three hiPSC lines were used in this study: Crt1[73], LiPSC-GR1.1[74] (kindly provided by W.-H. Zimmermann, IPT/UMG) and HipSci HPSI1013i-wuye_3[75] (ECACC 77650131, generated using the Cyto-Tune®-iPS Kit from Life Technologies and kindly provided by the Wellcome Trust Sanger Institute, UK). hiPSCs were maintained in TeSR-E8 medium (Stemcell, 05990) on Geltrex-coated cell culture dishes prior to differentiation. CM differentiation was induced by the temporal modulation of the Wnt pathway[76]. Wnt activation was initiated on day 0 using CHIR99021 (Hölzel, C-6556). After 48 h, Wnt inhibition was performed by treating cells with IWP2 (Sigma-Aldrich, 681671) for 48 h. For atrial cardiomyocyte specification, cells were stimulated with retinoic acid (Sigma-Aldrich, R2625) between day 3 and day 6 of differentiation[77,78]. A high-purity CM population was achieved using the metabolic selection method[79]. The cellular identity was regularly confirmed based on cell morphology and beating frequency[77] as well as using flow cytometry (Sony SH800). Staining of hiPSC-aCM and hiPSC-vCM using anti-cTNT antibodies (Miltenyi, REA400|1C11) revealed >90% of cTNT-positive CM after metabolic selection as previously described[77]. In case of atrial cardiomyocyte differentiation, ~90% of cells were identified as atrial CMs using antibodies targeting MLC2a (Miltenyi, REA398|56F5). In case of hiPSC-vCM about 90% of cells were marked by an antibody recognizing the ventricular CM marker MLC2v

(Miltenyi, REA401|330G5)[77]. Representative QC data is shown in Supplementary Fig. 18a–d.

## Generation of enhancer knockout hiPSCs cell lines

The ribonucleoprotein complex (RNP) was assembled by incubating ArciTect Cas9-eGFP Nuclease (Stem Cell Technologies, 76006) with custom-designed synthetic sgRNA (Genscript, listed in Supplementary Data 10) for 15 min at room temperature, following the manufacturer's protocol. Alt-R CRISPR-Cas9 Electroporation Enhancer (IDT, 1075916) was then added to the RNP complex.

hiPSCs were dissociated into a single-cell suspension using Accutase. Transfections were performed using the Neon transfection system (Thermo Fisher Scientific). A $5 \times 10^5$ hiPSCs cell pellet was resuspended in 15 μl Neon Buffer R and mixed with 15 μl RNP complex. Electroporation was carried out with two pulses at 1100 V for 30 ms. Cells were then plated onto Matrigel–coated plates and cultured in mTeSR Plus medium (Stem Cell Technologies, 100-0276) supplemented with CloneR2 (Stem Cell Technologies, 100-0691). Transfected cells were grown for 24–36 h.

Cells were then harvested using Accutase, resuspended in mTeSR Plus medium supplemented with CloneR2, and stained with 7-Aminoactinomycin D (7AAD) (VWR, CAYM11397) to assess viability. The cell suspension was filtered through 30 μM CellTrics filters (Sysmex, 04-004-2326). 7AAD-negative/GFP-positive single cells using a SH800S cell sorter (Sony Biotechnology) equipped with a 130-μm nozzle. Forward scatter width and height parameters were used to exclude doublets. Single cells were sorted into Matrigel–coated 96-well plates containing mTeSR Plus medium supplemented with CloneR2. After recovery and clonal expansion, genotyping of the clones was performed using Phire Tissue Direct PCR Master Mix kit (Thermo Scientific, F170). Primers used for PCR screening are listed in Supplementary Data 10.

## sgRNA cloning into lentiviral CRISPRi vector and lentivirus production

An expression plasmid was generated using a plasmid obtained as a gift from Charles Gersbach[35] (Addgene #71237, pLV hU6-sgRNA-CAG-dCas9-KRAB-T2a-GFP) in which a synthetically synthesized DNA fragment replaced the human U6 and ubiquitin C promoter. This 1105 bp DNA fragment consisted of the following components: Mouse U6 promoter, gRNA entry site, and CMV enhancer. For this exchange, we used the restriction enzymes PacI and XbaI. To clone sgRNAs (Supplementary Data 10) into the expression plasmid, two complementary oligonucleotides (ordered from eurofinsgenomics) were annealed and ligated into the expression vector via Golden Gate cloning with Esp3I. The ligation product was transformed into competent Escherichia coli cells, and plasmids of positive clones were isolated using the NucleoSpin Plasmid Mini kit (Macherey-Nagel). CRISPRi expression vectors containing sgRNAs, specific for one regulatory element, were pooled equimolarly. Together with envelope and helper plasmids (gifts from Didier Trono, Addgene: 12259, 12260), and were used for lentivirus production in HEK293T cells.

## sgRNA cloning into AAVi vector and AAV6 production

sgRNAs (listed in Supplementary Data 10) were cloned into AAVi plasmid encoding Staphylococcus aureus dCas9-KRAB (Addgene #214609)[34] using golden gate cloning with BbsI (NEB, R3539). The ligation product was then transformed into chemically competent Escherichia coli cells, and plasmids of positive clones were isolated using the NucleoSpin Plasmid Mini kit (Macherey-Nagel).

AAV6 production was carried out in AAVpro 293T cells (Takara, 632273), which were co-transfected with AAVi plasmid and AAV helper and packaging plasmid (rep2/cap6) using polyethylenimine (Sigma-Aldrich, 408727). Viral particles were purified using AAVpro extraction solution (Takara, 6235) following the manufacturer's protocol. The

purified AAV particles were concentrated using 100-kDa Amicon Ultra Centrifugal Filter Units (Sigma-Aldrich, UFC510096). The AAV titer was determined by qPCR using the AAVpro Titration Kit v2 (Takara, 6233).

## Transduction of hiPSC-CMs

For all experiments we used at least three independent hiPSC-CM differentiations. AAV- or lentiviral-mediated transductions were performed on hiPSC-CMs at post-differentiation day >50.

Lentiviruses were used to transduce hiPSC-CMs at a multiplicity of infection (MOI) of 3, facilitated by the addition of polybrene (Merck). GFP-positive lenti-transduced hiPSC-CMs were sorted into RLT buffer (Qiagen) after 12 days for gene expression analysis or into PBS containing 1% BSA for ATAC-seq and Hi-C analysis. In case of AAVi, transduction was performed at an MOI of 5000–30,000, and cells were collected after 7 days for subsequent analysis.

## RT-qPCR and RNA-seq

RNA was isolated with the RNeasy Micro Kit (Qiagen), including on-column DNase digestion. For RT-qPCR analysis RNA was transcribed into cDNA using the SuperScript™ IV First-Strand Synthesis System (Invitrogen, 18091300), and the 1x SsoAdvanced Universal SYBR® Green Supermix (Bio-Rad, 1725275) was used to determine the expression of the target genes using a Roche LightCycler® 480 Instrument II using primers ordered from eurofinsgenomics. The SMART-Seq Total RNA Pico Input (ZapR Mammalian, Takara, 634356) was used for the generation of RNA-seq libraries.

## Automated patch-clamp recordings

Inward-rectifier $K^+$ currents were recorded using SyncroPatch 384 (Nanion Technologies GmbH, Germany), as described previously[80,81]. hiPSC-vCMs were analyzed on thin borosilicate glass, single-aperture 384-well planar fixed-well APC chips (1xS-type NPC-384T). Inward rectifier potassium currents ($I_{K1}$) were measured by applying a ramp pulse from −100 to +40 mV (0.5 Hz).

The external KCl concentration of 20 mM was used to facilitate a positive shift in $I_{K1}$ reversal potential, which allows for a larger current acquisition of the inward component of $I_{K1}$ at −100 mV. $I_{K1}$ was identified as current responsive to $Ba^{2+}$ blockade (1 mM).

## Multi-electrode array (MEA) measurements

hiPSC-vCM were seeded at a density of 30,000 cells in 5 µl media onto Geltrex-coated recording electrodes of a CytoView MEA plate (Axion Biosystems) using the spot seeding method with PBS in the space between the wells to create a humid atmosphere preventing cells from drying out in the process. Six hours after seeding, 300 µl media was added to each well. Baseline field potential duration was recorded 2 days post-seeding using a Maestro detection system (Axion Biosystems). Cells were equilibrated for at least 30 min at 37 °C and 5% CO2 within the system prior to recording. Following baseline measurements, cells were transduced with AAV viral particles at MOI > 30,000 Vg/cell. Field potentials were recorded 7 days post-transduction. The AxIS Navigator (version 3.12.7) and the Cardiac Analysis Tool (3.3.3) were used to record and analyze the cardiac activity parameters of spontaneous action potentials.

## ATAC-Seq

For ATAC-seq we used the OMNI-ATAC protocol[82]. The transposition approach was scaled to cell/nuclei number (2.5 µL Tn5 Tagment DNA Enzyme 1 (Illumina) per 50,000 cells). Samples were amplified after the addition of EvaGreen, and the fluorescence was monitored during the amplification. Amplification was stopped when the fluorescent signal reached the inflection point.

## Sequencing of DNA libraries

The concentration of DNA libraries was determined by Qubit (Invitrogen), and the insert size using a Bioanalyzer or Tapestation (Agilent Technologies). Pooling of multiplexed sequencing samples, clustering, and sequencing were carried out on Nextseq2000, Novaseq (Illumina) or AVITI (Element Biosciences) sequencers. All libraries were sequenced in paired-end mode.

## Methyl-seq analysis

Raw reads were trimmed as paired-end reads using *TrimGalore!* (https://github.com/FelixKrueger/TrimGalore) with default parameters. Trimmed reads were aligned to human genome assembly hg19 using bwameth (Galaxy Version 0.2.2+galaxy1, paired-end mapping) and PCR duplicates removed using RmDup (Galaxy Version 2.0.1). CpG-methylation metrics were extracted using MethylDackel (Galaxy Version 0.3.0). For visualization of CpG methylation, values were smoothed using a centered running window of 3 CpGs.

MethylSeekR[31] was used to partition the genome. Partially methylated domains (PMDs) were identified after training on chromosome 1, and PMDs shorter than 100 kb were filtered out. Subsequently, low methylated regions (LMRs) and unmethylated regions (UMRs) were annotated using the parameters m.sel = 0.5 and fdr.cut-off = 10. Regions overlapping annotated transcription start sites were filtered since they are likely linked to gene body demethylation and do permit CRE annotation[11,83]. A replicate-based differential methylation analysis of CREs was conducted using Metilene[84]. A minimum mean methylation difference of 10% and an adjusted $p$ value ($q < 0.05$) were considered significant.

## RNA-seq data analysis

Published gene expression data sets were mapped to the human genome (hg19) using RNAStar[85] and PCR duplicates were removed using RmDup (Galaxy Version 2.0.1). Cufflinks[86] was used to calculate FPKM expression values. Reads were counted and Deseq2[87] was used to normalize count tables and identify differentially expressed genes using an FDR corrected $p$ value of 0.05.

## ATAC-Seq analysis

ATAC-Seq sequencing data were cleared of adapter sequences using *Cutadapt*[88]. We used *Bowtie2*[89] to assign the sequencing data to the corresponding position in the human genome hg19 in paired-end mode. The maximum fragment length was set to 1000 bp. Sequencing pairs pointing away from each other were allowed (--dovetail yes), and the sensitive (--very-sensitive) mode was used. PCR duplicates were removed using RmDup (Galaxy Version 2.0.1).

## ChIP-seq analysis

We used *Bowtie2*[89] to assign the sequencing data to the corresponding position in the human genome hg19 in paired-end mode and PCR duplicates removed using RmDup (Galaxy Version 2.0.1). Diffbind[90] was used to identify differential H3K27ac marking of target regions using a $p$ value cut-off of 0.05.

## Hi-C matrices preprocessing

The raw paired-end reads were trimmed using *TrimGalore!* (https://github.com/FelixKrueger/TrimGalore) with default parameters. The resulting trimmed reads were aligned as single-end reads to the human genome hg19 *Bowtie2*[89] BAM files of technical replicates were merged using *SAMtools* (v1.15.1)[91] and then processed using HiCExplorer (v3.7.2)[92].

We used *hicBuildMatrix* (parameters --binSize 1000 --restriction-Sequence GATC GAATC GATTC GAGTC GACTC --danglingSequence GATC AATC ATTC AGTC ACTC --minMappingQuality = 1) to generate Hi-C matrices (h5 format) at a 1 kb resolution. The Hi-C matrices of

biological replicates were merged with *hicSumMatrices*. The bin size was adjusted for different resolutions with *hicMergeMatrixBins*. *hicCorrectMatrix* was used to extract and filter out low-coverage regions. Quality control measures for Hi-C matrices were evaluated with *hicQC*, and the correlation between Hi-C matrices data was computed using *hicCorrelate* (parameters --method=pearson --zMin 0 --zMax 1 --plotNumbers). The distribution of long- and short-range contacts was calculated using *hicPlotDistVsCounts*.

## A/B compartment annotation

HOMER tools[93] were used for A/B compartment segmentation according to the documentation for all autosomes. A bin size of 50 kb was used with a window of 100 kb to calculate the first and second principal component (PC1 and PC2). The PC1 was used despite a few exceptions for chr4, 9, and 16, where PC2 corresponded to the A/B classification. The signs for PC1 and PC2 were flipped according to H3K27ac ChIP-seq data. A replicate-based differential A/B analysis was performed using HOMER tools with FDR correction, a $p$ value cut-off of 0.05, and an absolute $\log_2$(fold change) >1 to identify domains with dynamic A/B status.

## TAD identification

Hi-C matrices of 50 kb resolution were normalized using *hicNormalize* tool (--normalize smallest) of hicExplorer toolkit[92]. TADs were identified using *hicFindTADs* --thresholdComparisons 0.05). Average TAD-seperation scores were calculated using *multiBigwigSummary* (Galaxy Version 3.5.4+galaxy0) and Pearson correlation analysis was performed using *plotCorrelation* (Galaxy Version 3.5.4+galaxy0).

## Hi-C viewpoint analysis

We analyzed genome-wide viewpoints to identify promoter-interacting regions using merged Hi-C matrices for each group. First, Hi-C matrices of 1 kb resolution were scaled to the smallest sequencing depth using *hicNormalize*. Viewpoints were generated for all gene promoters listed in the RefSeqSelect hg19 NCBI annotation. Promoter regions were defined as TSS ± 2.5 kb. Viewpoints with low coverage were filtered out using *chicQualityControl* (parameters --fixateRange 200000 -s 0.06). A set of 10,000 random genomic regions equivalent to the bin size was used to calculate and build a continuous negative binomial distribution model (background model). To exclude functional promoter interactions from the background model random regions passing were selected from inactive genome regions and filtered using *chicQualityControl*. The background model was calculated using *chicViewpointBackgroundModel* (parameters --fixateRange 200000 --averageContactBin 5). Background model adjustments were made to circumvent chromatin interaction overestimation owing to the sparse background signal in regions distant from the viewpoint. Bins exceeding 200 kb from the viewpoint were replaced with average background metrics derived from the last ten bins within the 200 kb adjacent intervals. The interaction frequency of 2 Mb region around each viewpoint was assessed with *chicViewpoint* (parameters --averageContactBin 5 --range 1000000 1000000). Interaction domains were determined by merging bins with relative interaction frequencies 1.5 fold above the background and interaction $P$ values were recomputed. Regions not meeting the cut-off $P$ value were rejected (*chicSignificantInteractions* with --pValue 0.0000000001 --range 1000000 1000000 --truncateZeroPvalues --fixateRange 200000 --xf 1.5 --peakInteractionsThreshold 10). The selected $P$ value cut-off corresponds to a Bonferroni-corrected FDR $P$ value of 0.005. Interaction domains adjacent to the center of each viewpoint (10 kb window) were excluded.

## Visualization Hi-C plots

DeepTools[94] and pyGenomeTracks[94] integrated into the Galaxy platform[95] were used to plot Hi-C heatmaps, viewpoints, RNA-seq, ChIP-seq, and ATAC-seq. For Hi-C, viewpoint analysis tracks were exported using *chicExportData* and *heatmaps* were generated from ICE-corrected matrices of 5 kb bins.

## Differential interaction analysis

We employed *chicAggregateStatistic* and *chicDifferentialTest* tools to identify potential differential promoter interactions between groups using merged Hi-C matrices (NF-LV-CM versus NF-LA-CM and NF-LV-CM versus F-LV-CM) with a significance level of 0.05 (chi-square test). A replicate post-hoc analysis was performed to ensure greater robustness. Therefore, we extracted the raw interaction data for the potential differential interactions and calculated the $\log_2$(fold change) and corresponding $P$ value of the relative interactions ($t$-test). A $P$ value of 0.05 was used as a cut-off criterion. The replicate-based analysis requires reproducibly covered regions. Therefore, we filtered out regions with a coverage of less than 1000 contacts per group. The gene expression was extracted for the obtained differentially interacting regions, and the fold change of gene expression was computed.

## GWAS and eQTL analysis

We obtained genetic variants associated with human disease from the GWAS catalog[49] (downloaded on May 30, 2023) and original studies[50,51]. We employed the Genomic Regulatory Elements and GWAS Overlap tool (GREGOR)[96] to assess the enrichment of variants within specified regulatory elements while accounting for the structure of linkage disequilibrium (LD). Using GREGOR, proxy variants were identified in LD ($r^2 > 0.7$) for each lead variant within a window of 1 Mb. LD scores were retrieved from the 1000 Genome project for the European population[97]. For statistical analysis of variants in target gene regions, the observed overlap of lead or proxy variants was compared to the expected overlap calculated using a random set of matched control variants. We assessed the enrichment based on the ratio of observed to expected overlaps. The significance of this enrichment was ascertained by assuming a sum of binomial distributions as implemented in GREGOR. A $P$ value of less than 0.05 was considered significant.

The eQTL data used for this manuscript were downloaded from the GTEx Portal[52] (V10) on 12/16/2024 (dbGaP accession number phs000424.vN.pN). The portal was used to plot gene-variant pairs.

## Gene ontology (GO) and Genomic Regions Enrichment of Annotations Tool (GREAT)

Gene Ontology network analysis was conducted using the Cytoscape plugin ClueGO (v2.5.10)[98]. The analysis was based on the following criteria: Min GO Level = 4, Max GO Level = 8, Min number of Genes = 10, and Min Percentage = 8.0. Bonferroni step-down correction was applied to $P$ values to control for multiple testing. The GO term connectivity threshold was set to 0.3 (kappa value), and only pathways with a $P \leq 0.01$ were considered significant. The resulting terms were grouped by function, and the leading term in each group was selected based on its significance. The findings were then illustrated using Cytoscape (v3.10.1).

Peak set gene annotation was done using GREAT (version 4.0.4)[43,99] with the following parameters: Basal extension: 5000 bp upstream, 1000 bp downstream, 1,000,000 bp max extension, curated regulatory domains included.

## MOTIF prediction

We performed de novo and known motif analysis using Homer[93]. We used TOBIAS TFBScan (v0.13.3)[100,101] to call CTCF motifs (Jaspar: MA0139.1)[102] in a set of peaks.

## External data

RNA-seq data for NF-LA-CM was obtained from Ouwerkerk et al.[103], and for NF-LV-CM, F-LV-CM from Gilsbach et al.[11]. ChIP-seq and DNA methylation data for F-LV-CM and NF-LV-CM were derived from

Gilsbach et al.[11]. ATAC-seq data sets of NF-LV-CM and NF-LA-CM were derived from Hocker et al.[19]. Ventricular CTCF ChIP-seq (ENCFF053 RDI) and DNA methylation data were obtained from the ENCODE portal[104]. All external data sets were reanalyzed starting from raw sequencing reads, apart from single-nucleus ATAC data, for which mapped data were used.

## Reporting summary

Further information on research design is available in the Nature Portfolio Reporting Summary linked to this article.

## Data availability

Genetic variants were removed from raw sequencing reads using BAMboozle[105] to de-identify genomic data. The resulting sequencing reads have been deposited in the NCBI Sequence Read Archive (SRA) under the accession code PRJNA1348139. In compliance with the ethical approval, patient history and data are unassigned. Access to genetic information is available on request. Source data are provided with this paper.

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

## Acknowledgements

We thank Christian Braun (Forensic Institute, Ludwig-Maximilians-University, Munich, Germany) for providing forensic samples of cardiac tissue. We thank the members of the Freiburg CardioVascular BioBank (CVBB) and the Cardiovascular Biobank (KaBi-DHM) of the TUM University Hospital German Heart Center, Technical University Munich, as well as the cardiovascular surgeons of the University Heart Center Freiburg-Bad Krozingen and the TUM University Hospital German Heart Center, Technical University of Munich for support during tissue collection and processing. We are very grateful to the patients who donated tissue to this research. We thank Katalin Pálfi for excellent technical assistance. We thank Prof. Dr. Bernhard Brüne (Pathobiochemistry Goethe-University, Frankfurt am Main, Germany) for access to Next-Seq2000 and the European Molecular Biology Laboratory GeneCore (EMBL, Heidelberg, Germany) as well as the Max Planck Institute of Immunobiology and Epigenetics Deep Sequencing Facility (Freiburg, Germany) for sequencing services. This study was supported by the German Research Foundation (DFG) Collaborative Research Centers 1425 (project-ID 422681845, projects P02 and S03 to R.G.) and 1550 (project-ID 464424253, project A02 to R.G.), the Baden-Württemberg Stiftung GmbH (ID13 Human-CM-3D-Genome to R.G.), and the DFG projects 558598989 and 386460455 (to R.G.). DFG to N.V. (VO 1568/3-1, VO 1568/3-2, VO1568/4-1, VO1568/6-1) and under Germany's Excellence Strategy—EXC 2067/1—390729940). This work was supported by the German Center for Cardiovascular Research (DZHK) (project-ID 81X24300110 to R.G. and K.S.B., and project-ID 81X4500124 to R.G. and C.S.). The authors acknowledge the support of the Freiburg Galaxy Team, Rolf Backofen (Bioinformatics, University of Freiburg, Germany), funded by the German Federal Ministry of Education and Research (grant 031 A538A de.NBI-RBC) and CRC1425. The authors acknowledge the data storage service SDS@hd, supported by the Ministry of Science, Research and Arts Baden-Württemberg and the DFG through grant INST 35/1503-1 FUGG. The Genotype-Tissue Expression (GTEx) Project was supported by the Common Fund of the Office of the Director of the National Institutes of Health, and by NCI, NHGRI, NHLBI, NIDA, NIMH, and NINDS. IS was partially funded by myriamed GmbH. Myriamed GmbH had no influence on the study design, study conduct and data interpretation. We thank W.-H. Zimmermann (IPT/UMG) for providing us with the LiPSC-GR1.1 line. Generation of LiPSC-GR1.1 (also referred to as TC1133 or RUCDRi002-A; lot number 50-001-21) was supported by the NIH Common Fund Regenerative Medicine Program[74]. The NIH Common Fund and the National Center for Advancing Translational Sciences (NCATS) are joint stewards of the LiPSC-GR1.1 resource. Repairon GmbH acquired and imported a vial of the TC1133 master cell bank, from which a Working Cell Bank (WCB) was created. Myriamed GmbH acquired a derivative of the WBC from Repairon GmbH and provided a non-GMP derivative thereof to the Institute of Pharmacology and Toxicology (IPT) at the University Medical Center Göttingen (UMG) for non-commercial research use. We acknowledge financial support from Heidelberg University for the publication fee.

## Author contributions

R.G., S.H., R.B., and P.L. designed research; S.H., R.B., P.L., N.D.P., K.S.B., M.D., T.L., S.K., I.S., and R.G. performed research; C.S., S.H., R.B., P.V., B.A.G., M.K., H.L., P.L., N.D.P., I.S., N.V., and R.G. analyzed data; and R.G., S.H., R.B., and P.L. wrote the paper.

## Funding

## Competing interests

The authors declare no competing interests.

## Additional information

[1]Institute of Experimental Cardiology, Medical Faculty Heidelberg, Heidelberg University, Heidelberg, Germany. [2]DZHK (German Center for Cardiovascular Research), partner site Heidelberg/Mannheim, Heidelberg, Germany. [3]Institute for Cardiovascular Physiology, Goethe University Frankfurt, Frankfurt, Germany. [4]Institute of Pharmacology and Toxicology, University Medical Center Göttingen, Göttingen, Germany. [5]Cluster of Excellence "Multiscale Bioimaging: From Molecular Machines to Networks of Excitable Cells", Georg-August-University Göttingen, Göttingen, Germany. [6]DZHK (German Center for Cardiovascular Research), partner site Göttingen, Göttingen, Germany. [7]Bioinformatics Group, Department of Computer Science, University of Freiburg, Freiburg, Germany. [8]Department of Cardiovascular Surgery, Institute Insure, TUM University Hospital German Heart Center, School of Medicine and Health, Technical University of Munich, Munich, Germany. [9]DZHK (German Center for Cardiovascular Research), partner site Munich Heart Alliance, Munich, Germany. [10]Department of Cardiology, Angiology and Pneumology, University Hospital Heidelberg, Heidelberg, Germany. [11]Department of Cardiology and Pneumology, University Medical Center, Göttingen, Germany. [12]Institute of Pharmacology and Toxicology, University of Würzburg, Würzburg, Germany. [13]Institute of Clinical and Experimental Pharmacology and Toxicology, University of Freiburg, Freiburg, Germany.
✉e-mail: Ralf.Gilsbach@uni-heidelberg.de

