## [Transparent Peer Review file · Nature Communications]

Chamber-Specific Chromatin Architecture Guides Functional Interpretation of Disease-Associated Cis-Regulatory Elements in Human Cardiomyocytes

Corresponding Author: Professor Ralf Gilsbach

Version 0:

Reviewer comments:

Reviewer #1

(Remarks to the Author)

Haydar et al. provide Hi-C data for purified CMs from human non-failing LV and RA as well as from failing LV and combine this with their previously published epigenomic and gene expression data and use this to link chamber-specific chromatin interactions to gene expression programs and perform overlaps with variants from different diseases/traits. Overall, the Hi-C data is of good quality but limited to only cardiomyocytes with regards to cell type specificity. Though the finding of chamber-specific chromatin interactions is interesting, no evidence is provided to validate that any of the identified chamber-specific PIDs regulate chamber-specific gene expression. Furthermore, the authors validate CREs for two loci that are associated with QT duration, which given the authors' identification of disease-specific PIDs, seems odd and not consistent with the studies on heart failure samples. Finally, the work is somewhat incremental to recent epigenomic and snATAC-seq studies on human hearts and/or cardiomyocytes (Zhang et al., PMID: 31427791, Anene-Nzeiu et al. PMID: 32866060, Avantis et al. PMID: 32111823, Foo studies PMID, Hocker et al. PMID: 33990324, Spurrel et al. PMID: 36130500 and Ameen et al. PMID: 36563664). As a result, the findings are somewhat descriptive and incremental to what has already been reported, thus limiting the impact and relevance of the findings. Overall, the manuscript in its current iteration has a number of issues that raise concerns about its suitability for Nature communication.

Major:

- 1) Can the authors provide evidence that any of the identified chamber-specific PIDs drive chamber-specific gene expression, for example using transgenic reporter assays in mice?
- 2) Why did the authors focus on QT-interval-associated CREs rather than HF-associated CREs for validation given that they identified PIDs that are specific to either failing or non-failing LV CMs? Were there no overlaps with HF variants for any of these PIDs? The lack of findings validating HF variants limits the relevance and significance of the findings.
- 3) As changes in PIDs seem to account only for a subset of the gene expression changes, are there any changes in TADs/compartments between the different samples that could explain some of the gene expression differences?
- 4) Fig. 3d-f: The increased interaction of the S-E with the NPPA promoter in F-LV-CMs is interesting but looks very subtle. Can the authors verify this interaction using a more quantitative method?
- 5) Fig. 5: The authors should provide a table with target genes for the GWAS hits?
- 6) The authors should compare findings to other recent human heart epigenomic and snATAC-seq studies by Zhang et al., PMID: 31427791, Anene-Nzeiu et al. PMID: 32866060, Avantis et al. PMID: 32111823, Foo studies PMID, Hocker et al. PMID: 33990324, Spurrel et al. PMID: 36130500 and Ameen et al. PMID: 36563664.
- 7) Author should show that hiPSC cardiomyocyte model also display similar epigenetic landscape and chromatin interactions as human ventricular cardiomyocytes to show that hiPSC cardiomyocytes could be used as a model to test studied CREs. This is especially important as the authors have shown that there are significant differences in both epigenomic and chromatin landscape between different types of cardiomyocytes.
- 8) Knockouts of CREs and knockin of genetic variants should be performed to validate CRISPRi studies.
- 9) Functional electrophysiologic studies should also be performed in hiPSC cardiomyocytes to confirm the impact of perturbing CREs.
- 10) It is somewhat surprising that perturbing AR2 CRE within the SCN5a locus has no functional impact on SCN5a expression when the authors' data and analyses suggest that it should, thus raising concerns about the utility of the authors' findings.

Minor

- 1) Better figure labeling: e.g. for some figure panels one has to look at the figure caption to know what the different colors mean.
- 2) Line 136-139: Define PID better (how is this different from a loop?). PIDs span 54 MB and overlap with 30k CREs spanning 24 MB. What is the remaining 30 MB if not CREs?
- 3) Lines 161-165: This section is difficult to follow. It's not clear how can you have diminished CTCF-mediated chromatin interactions without impacting higher-order chromatin organization? Are these CTCF sites not at TAD boundaries?
- 4) Supplementary Fig 11: There's a clear difference for NF-LA LMRs, but very marginal for NF-LV and F-LV. Can the authors do statistics for these comparisons? Furthermore, it could be interesting to plot ChIP-data, in particular H3K27ac, for the LMRs, as there maybe differences in enhancer activity that are not captured by CpG methylation.

Reviewer #2

(Remarks to the Author)

In this MS, the authors employ Hi-C to analyse chromatin interactions in healthy atrial and ventricular cardiomyocytes as well as failing ventricular CM. By also performing this analysis in ventricular tissue – which comprises multiple cell types, they illustrate that analysis specifically of the CM compartment is required to gain insight into how chromatin organisation can relate to other epigenetic marks and transcription in CM. This requirement to specifically analyse CM and not tissue builds on their elegant studies on cardiac epigenetics. In this current MS, the authors demonstrate chamber specific chromatin interactions that were related to associated transcription and transcription factor binding. Analysis of locations of SNP associated with cardiac disease determined these to be localised to chamber specific chromatin interacting regions. However, these did not appear to change with disease. Interestingly CRISPRi disruption of sites of interaction with the KCNJ1 locus that coincided with identified SNPs, decreased KCNJ1 expression in iPS-CM suggesting functional importance.

This MS provides important new information regarding the regulation of gene expression across the chambers of the heart and how it may be affected by SNPs located at cis regulatory elements. Only through correlating these chromatin interactions specifically within CM and establishing the functional role of the CREs identified could these findings be reached.

There are some questions which require addressing:

- 1- In Fig. 2d, only two groups are being compared, but the Kruskal-Wallis test was used. Since the Kruskal-Wallis test is generally used for comparing more than two groups, shouldn't the Wilcoxon rank-sum test be used instead?
- 2- In Fig.3, the H3K27ac-marked super-enhancer is located upstream of both NPPA and NPPB, but the focus was primarily on NPPA, despite the RNA expression data showing that NPPB was also affected. Was this due to interactions of CRE only with the NPPA promoter? According to the Human cell atlas, both NPPA and NPPB are expressed in atrial cardiomyocytes, but NPPB is also expressed in some ventricular CM. Wouldn't it be valuable to discuss NPPB in more detail in the context of ventricular CM as well? It seems that the RNA peaks for NPPA and NPPB do not fully align with the gene locus. Perhaps the figure needs adjusting.
- 3- In Fig. 5, analysis was restricted to promoter interacting CREs in non failing atrial and ventricular CM. This analysis identified enrichment for disease associated SNPs in these promoter interacting CRE that were related to disease in the related chamber. Would a stronger relationship with disease related SNPs be observed in PI-CRE in failing CMs.
- 4- Similarly, it would be informative and add weight to the relevance of the interaction, if RNA levels, ATAC peaks, H3K27ac peaks were provided for KCNJ2 for failing as well as non-failing LV CM samples in Fig. 6a.
- 5- In Fig 6b, the silencing of KCNJ2 promoter-interacting elements 3 and 4 significantly reduced KCNJ2 gene expression, yet Fig 6e shows no significant effect on the interaction frequency between these regions and the KCNJ2 promoter. Supplementary Fig 4 suggests there are only four differentially promoter-interacting domains (PIDs) for KCNJ2, possibly including elements 3 and 4. How can the silencing of interacting elements 3 and 4 reduce gene expression without affecting the promoter interaction frequency? What might explain this apparent disconnect between gene expression and promoter interaction?
- 6- Low IK1 is a feature of iPS-CM contributing to their immature AP. Given that KCNJ1 is the contributing channel to this current, would its expression not also be basally very low in iPS-CM. The authors show relative change in expression of KCNJ1 following CRISPRi. The relevance of these changes may however not be functionally important if expression is very low and may also overestimate the effect of CRISPR – i.e. CRISPR does not need to be very effective to reduce KCNJ1 expression. Could the levels of mRNA expression be shown and how do these compare to levels in adult vCM.

Reviewer #3

(Remarks to the Author)

In "Linking Chamber-Specific Spatial Chromatin Interactions to Disease Variants and Gene Programs in Human Cardiomyocytes" Haydar et al. conduct genome wide chromatin interaction mapping in ventricular vs atrial cardiomyocytes and ventricular cardiomyocytes from failing left ventricles. They then conducted multiple analyses related to cis regulatory element – promoter interactions and discovered chamber specific interaction that correlated with specific gene expression. Moreover, chamber specific interactions were more likely to be found in regions of disease associated genetic variants – e.g. AFIB associated variants linked with cis regulatory elements enriched in the atria vs ventricle. Additional CRISPRi experiments validated the importance of these promoter-cis regulatory element interaction on target gene expression. While this is not the first manuscript to associate genetic variants with cis regulatory elements, it is the first to show chamber specific cis regulatory elements and links to variants of chamber specific disease etiology. This work will impact workflows for investigating/validating genetic variants. These points are sufficiently impactful to indicate this will be a worthwhile addition to the literature and of appropriate scope for consideration by Nature Communications.

The biggest surprises in the data for this reviewer were that 1) largely the "...interactions were not affected in CMs of failing hearts and not linked to pathological gene expression on a genome wide scale." And 2) CRISPRi of CREs failed to disrupt interactions despite impact on gene expression and chromatin accessibility (ATAC).

These findings could have profound implications for current models/understanding of gene expression regulation in the heart. Are the authors confident that the techniques used and data generated were sufficiently robust to draw these conclusions (positive control for 6e would help)? If so, please provide direct language in rebuttal explaining why and direct language in manuscript highlighting the implication of the findings.

Minor issues indicated below:

"Histone code" bar in multiple figures appears inaccurate in terms of RNA production.

Sentence from line 271-273 missing at least one word.

What is the interpretation of interaction domain size difference between CM and Tissue 2C

Please articulate potential limitations of comparing current HiC to epigenetics data from another study.

"A recent study shows that enhancer promoter interactions are becoming more instructive during muscle terminal differentiation." Instructive to who/what?

Labeling in some figures could be improved for interpretability – Suggest adding color code for groups to individual figure panels where appropriate.

While the methods section is generally excellent, this reviewer has some confusion regarding the GWAS and variant vs proxy variant identification using a 1Mb window. For the proxy variants found at extreme distances shown to overlap CRE, was that CRE associated with a promoter nearer the lead variant? Comment may be outside of scope.

As indicated introduction contains a sentence indicating "interactions were not affected in CMs of failing hearts and not linked to pathological gene expression on a genome wide scale" yet discussion contains sentence indicating "Our data suggests that CTCF-mediated chromatin interactions are affected in terminal heart failure in CM" then goes on to discuss NPPA regulation. The "our data..." sentence should be qualified in some way to indicate lack of support for this observation in the genome-wide analysis.

Version 1:

Reviewer comments:

Reviewer #2

(Remarks to the Author)

In response to the reviewers' comments, the authors have responded with a substantial revision of their MS including new data.

I am generally satisfied with the responses to my questions.

One remaining issue – I, as well as other reviewers comment on the choice of KCNJ2 (and its enhancers) as a target that is followed up upon with functional assays. This is followed up on as notably, the authors do not show any HF related interactions – perhaps owing to insufficient sample number.

Notably, experiments were performed to knockdown expression of a transcription factors and NPPA but consequences for hypertrophic remodelling was not performed.

Given this less related analysis of KCNJ2 to HF and link to LQT rather than HF related genes, I would modify the Intro and discussion.

Sudden cardiac death is commonly associated with HF as are other arrhythmic disease, which cardiomyocytes may be predisposed to under conditions of altered KCNJ2. This link to arrhythmia should be made rather than solely focus on HF.

But – this arrhythmia may also be a consequence of hypertrophic remodeling. It may also be argued that a longer APD could signal induction of a hypertrophic response. This could be discussed.

Reviewer #3

(Remarks to the Author)

Authors have addressed my concerns. Will be a solid addition to the literature.

Reviewer #4

(Remarks to the Author)

This study employs advanced multi-omics techniques (Hi-C, Methyl-seq, ChIP-Seq, ATAC-seq) to comprehensively investigate the epigenomics and 3D chromatin architecture of healthy human atrial and ventricular cardiomyocytes (CMs) as well as failing ventricular CMs. The authors functionally validated promoter-interacting CREs of the MYL2, MYH7, KCNH2, and SCN5A, and prototypic KCNJ2 genes in hiPSC-CMs using CRISPR interference (CRISPRi) approaches. These multi-omics efforts reflect tremendous work, and the additional data provided to address previous reviewer comments have strengthened the manuscript. However, several key issues still need to be addressed before publication.

Major

1. Unclear central goal: The current manuscript doesn't clearly articulate the central goal, hypothesis, or main research question. This lack of clarity makes it difficult to follow the purpose of the individual sub-sections in the Results. Please revise the abstract, Introduction and Results sections to clearly define the study's overarching objective.
2. Value added by chamber-specific Hi-C: From the title, it appears the authors intend to demonstrate that chamber-specific chromatin architecture revealed by Hi-C provides unique insights for functional interpreting disease associated Cis-regulatory elements. If this is correct, it would be helpful to explicitly summarize key discoveries that were uniquely revealed by chromatin Hi-C sequencing, but not by other epigenetic or transcriptomic profiling.
3. Hi-C resolution concern: On page 5 line 151, the manuscript states that "principal component analysis of the Hi-C contact matrices revealed minimal differences in chromatin compartmentalization among groups", and the author conclude that "both TAD structure and A/B compartmentalization are highly conserved across CM subtypes despite associated transcriptional and epigenetic changes" (line 157). However, considering that a 1 kb bin size was used for Hi-C matrices preprocessing, but larger bin sizes (50kb for A/B compartment and 100kb for TAD) were used for downstream analyses, this raises the possibility that the chosen resolutions may not be sufficient to detect chromatin architecture changes in diseased hearts. Please justify the chosen resolutions and discuss their limitations.
4. Suitability of hiPSC-CMs for functional validation: hiPSC-CMs were used for functional validation experiments. Since the authors acknowledge the maturation limitations of hiPSC-CMs, it would strengthen the study to compare the 3D chromatin architecture of primary CMs and hiPSC-CMs, particular at the loci studied, to assess whether hiPSC-CMs are a suitable model for those chromatin architecture functional assays.
5. This might be out of the central scope of this manuscript. But to support the conclusion that "CRISPRi precisely silences individual regulatory elements", an additional CRISPRi group targeting region 3 should be included in Fig. 8d.
6. The action potential duration (APD) varies significantly due to difference in ion channel expressions among individual hiPSC-CMs. The sample size (N = 6) in Fig. 8f is too small to draw a strong conclusion. It would be helpful to include CRISPRi target region 1 (promoter), which completely silenced gene expression (Fig.8e), as a positive control group in Fig.8f. In addition, patch clamp recordings using SyncroPatch 384 might be more sensitive than MEA for detecting APD changes and should be considered.

Minor

1. Please provide justification for why only male cardiac biopsies were used in this study.
2. Lin767 states: "hiPSC-vCM were seeded at a density of 30,000 cells in 5 μ l media onto Geltrex-coated recording electrodes of a 48-well CytoView MEA plate (Axion Biosystems) using the spot seeding method. Six hours after seeding, 300 μ l media was added to each well." It seems unrealistic that 5 μ l of media would adequately support cell viability for 6 hours in a 48-well-plate well. Please clarify this and ensure a reproducible protocol.

Reviewer #5

(Remarks to the Author)

In this study, the authors investigate chamber-specific cis-regulatory elements (PID-CREs) that interact with their target genes using good quality Hi-C data. They identify PID-CREs that are specific to atrial and ventricular tissues and demonstrate their role in regulating chamber-specific marker genes by CRISPRi. Furthermore, the study highlights an enrichment of SNPs within these regulatory elements that are associated with arrhythmia-related GWAS signals, including those linked to long QT syndrome, thereby providing a valuable framework for investigating and functionally validating disease-associated genetic variants.

The authors have carefully and thoroughly addressed all of Reviewer 1's comments. The revisions prompted by these comments have significantly strengthened the manuscript. Notably, the authors implemented several key improvements that enhance the robustness and functional relevance of their findings:

As recommended by Reviewer 1, the authors performed CRISPR interference (CRISPRi) targeting chamber-specific PID-CREs in atrial and ventricular hiPSC-derived cardiomyocytes (hiPSC-CMs), which led to changes in the expression of target genes.

They validated one of the CRISPRi results by performing complementary CRE deletion experiments, which yielded consistent outcomes.

In addition, the authors incorporated new functional data to support the impact of CRE perturbation in iPSC-CMs. Specifically, they conducted electrophysiological experiments, presented in Figure 8, which functionally validated two CREs at the electrophysiological level.

In their response, the authors also referenced recently published studies that corroborate their findings, particularly regarding the presence and relevance of PID-CREs.

Minor comment:

In their CRISPRi experiments, the authors used atrial and ventricular cardiac differentiation hiPSC. It would be helpful to report the enrichment of atrial and ventricular cardiomyocytes obtained from each respective differentiation, to provide better context for the interpretation of the functional perturbation results.

Overall, I find the authors' responses to be adequate, and the revised manuscript acceptable for publication in its current form. I therefore recommend acceptance.

Version 2:

Reviewer comments:

Reviewer #2

(Remarks to the Author)

The authors have made significant efforts to address the reviewers' comments. I am satisfied with the answers to my comments and the modifications to the MS.

I have no remaining issues.

Reviewer #4

(Remarks to the Author)

The authors have addressed my concerns.

Review summary:

We have addressed the reviewers' comments through a series of targeted experiments that confirm the robustness and functional relevance of our findings.

Key experiments and findings are summarized below:

- 1) We **cross-validated CRISPR silencing (CRISPRi) of cis-regulatory elements (CRE)** by combining CRE deletion experiments and CRISPRi-mediated silencing of published and newly identified CREs. (Supplementary Fig. 14a-c; Supplementary Fig. 21).
- 2) We demonstrate **that CRE-CRISPRi has no significant impact on promoter-CRE interaction strength**. Thus, it allows functional interrogation of CREs without a major impact on chromatin architecture, which is known to happen upon CRISPR deletions (Supplementary Fig. 14d-e).
- 3) We **perturbed two CREs associated with heart failure**, confirming their functional relevance (Fig. 5f,g; Fig. 4c + Fig. 6e).
- 4) We **validated chamber-specific promoter-interacting CRE** using complementary approaches:
 - eQTL analysis confirms the chamber-specific linkage of CREs and target genes (Fig. 7h-j).
 - In vivo enhancer reporter experiments from the VISTA database prove the chamber-specific activity of PIDs (Fig. 3d,e).
 - Perturbation of two CREs using CRISPRi shows the functional relevance of chamber-specific CREs for TF expression (Fig. 4).
- 5) We **integrate two GWAS studies to confirm the usability** of this resource and show that it **allows the stratification of sub-threshold genetic variants** (Fig. 7c-g).
- 6) We **functionally validated two CREs on the electrophysiological level** (KCNJ2 currents and repolarization time), providing mechanistic insights into how disease-associated CREs modulate electrophysiological properties implicated in the onset of ventricular arrhythmia (QT syndrome). One of the CREs carries a high-confidence GWAS QT syndrome variant. The second one carries a sub-threshold variant, supporting our data's usability for identifying novel risk loci. (Fig. 8e,f; Supplementary Fig. 20b-g).

We display and integrate these and additional new experimental data and analysis in the following figures:

- *Main Figures (23 new panels):* 3d,e; 4a-d; 5 a,d-f; 6c-e; 7c-j; 8 e,f
- *Supplementary Figures (31 new panels):* 11a-f; 12a-d; 13f,g; 14a-f; 19, 20a-g; 21a-e
- *Supplementary Tables (3 new tables):* 4-6
- *Reviewer Figures (7 new panels):* 1, 2a-d, 3a,b

REVIEWER COMMENTS

Reviewer #1 (Remarks to the Author):

Haydar et al. provide Hi-C data for purified CMs from human non-failing LV and RA as well as from failing LV and combine this with their previously published epigenomic and gene expression data and use this to link chamber-specific chromatin interactions to gene expression programs and perform overlaps with variants from different diseases/traits. Overall, the Hi-C data is of good quality but limited to only cardiomyocytes with regards to cell type specificity. Though the finding of chamber-specific chromatin interactions is interesting, no evidence is provided to validate that any of the identified chamber-specific PIDs regulate chamber-specific gene expression. Furthermore, the authors validate CREs for two loci that are associated with QT duration, which given the authors' identification of disease-specific PIDs, seems odd and not consistent with the studies on heart failure samples. Finally, the work is somewhat incremental to recent epigenomic and snATAC-seq studies on human hearts and/or cardiomyocytes (Zhang et al., PMID: 31427791, Anene-Nzeiu et al. PMID: 32866060, Avantis et al. PMID: 32111823, Foo studies PMID, Hocker et al. PMID: 33990324, Spurrel et al. PMID: 36130500 and Ameen et al. PMID: 36563664). As a result, the findings are somewhat descriptive and incremental to what has already been reported, thus limiting the impact and relevance of the findings. Overall, the manuscript in its current iteration has a number of issues that raise concerns about its suitability for Nature communication.

Response:

Thanks for reviewing the manuscript and suggesting experiments to increase its impact. As highlighted in detail in our responses to your points, we added substantial new experimental data and integrated published studies into our manuscript (as summarized on page 1). These experiments reinforce the biological and functional significance of our data on multiple layers. We believe that these updates markedly enhance the novelty and translational relevance of our study.

Major:

1) Can the authors provide evidence that any of the identified chamber-specific PIDs drive chamber-specific gene expression, for example using transgenic reporter assays in mice?

Response:

We thank the reviewer for the important question. We addressed this question using complementary in vitro and in vivo approaches as detailed below.

The first evidence comes from transgenic reporter experiments recently published by Kosicki et al. NAR 2024¹. Out of 380 enhancers active in embryonic mouse hearts, more than 50% overlapped with adult CM PIDs. These enhancers were related to cardiac genes. Notably, 14% overlapped with chamber-specific PIDs identified in adult NF-LV-CMs. A major fraction showed concordant chamber-specific reporter activity and promoter-interactions, providing in vivo evidence that chamber-specific PIDs can confer chamber-specific activity.

In contrast, VISTA-positive enhancers that did not overlap with adult CM PIDs were enriched for genes involved in general or early developmental programs, suggesting that they may be active in embryonic but not adult heart (Fig. 3d,e).

Furthermore, we perturbed chamber-specific PIDs in atrial or ventricular hiPSC-CM and could confirm the predicted regulatory activity. Silencing the ventricular-specific HEY2 PI-

CRE in hiPSC-vCM reduced HEY2 expression and silencing atrial-specific PI-CREs associated with NPPA/NPPB and TBX5 in hiPSC-aCM resulted in significant reduction of their respective target genes (Fig. 4,5). These results demonstrate the functional contribution of these chamber-specific CREs to chamber-restricted gene expression.

By integrating eQTL data² from human atrial appendages and left ventricle tissues, we identified genetic variants within chamber-specific PIDs impacting gene expression in a chamber-specific manner. Notable examples include CRE variants affecting NOS1AP and KCNK3 expression (Fig. 7 h-j).

Together, these additional lines of evidence support the conclusion that chamber-specific PIDs are an essential layer of gene regulation in human CM.

We updated the results sections accordingly (lines 215-228, 254-256, and 322-325) to clarify and strengthen this point. We also updated the methods section (lines 729-739, 747-748, and 911-913).

2) Why did the authors focus on QT-interval-associated CREs rather than HF-associated CREs for validation given that they identified PIDs that are specific to either failing or non-failing LV CMs? Were there no overlaps with HF variants for any of these PIDs? The lack of findings validating HF variants limits the relevance and significance of the findings.

Response:

We thank the reviewer for raising this point. Considering that disease-associated genetic variants are enriched in HF-associated CREs would imply that the genetic burden is modified by the onset of heart disease. We performed a GWAS analysis for HF-associated CREs and did not find a clear enrichment of genetic variants for any tested genetic trait. This suggests that HF-associated CREs are not modulating genetic disease risk factors in human LV-CM. In contrast, a strong enrichment for QT-associated variants was observed specifically in NF-LV-CM, driving our choice to focus on these genetic risk factors in this manuscript. The results of our GWAS analysis are summarized in Reviewer Fig. 1, which is attached below.

Reviewer Fig. 1: Heatmap displaying the significance of the enrichment calculated for genome-wide association study (GWAS) variants compared to random variants for NF-LV-CM PI-CREs and differential PI-CREs in F-LV-CM as compared to NF-LV-CM. The results for various organ disease traits from the NHGRI-EBI GWAS catalog are shown.

To further highlight HF-associated CREs, we added a new main figure to the manuscript (Fig. 6) showing promoter rewiring, H3K27ac-dynamics, and associated gene expression changes in F-LV-CM. We could not identify a suitable in vitro model to test the impact of rewiring or marking of CRE in heart disease. However, we silenced two candidate CREs using CRISPRi in hiPSC-CM that were picked based on either gained promoter interaction strength (NPPA) or lost H3K27ac-marking (HEY2) in F-LV-CM as compared to NF-LV-CM. Silencing of either CRE reduced expression of their respective target gene, indicating the

functionality of these HF-associated CREs (Fig. 4,5). We updated the results sections accordingly (lines 254-256 and 273-282).

3) As changes in PIDs seem to account only for a subset of the gene expression changes, are there any changes in TADs/compartments between the different samples that could explain some of the gene expression differences?

Response:

To address this question, we performed a comparative analysis of TADs and A/B chromatin compartments across all CM datasets. TADs are highly conserved across all CM groups. TAD insulation scores were strongly correlated, and TAD boundary strength metrics showed minimal variation between NF-LV-CM, NF-LA-CM, and F-LV-CM (Supplementary Fig. 11). Similarly, A/B compartmentalization was largely stable. Principal component analysis of the Hi-C contact matrices revealed only limited A/B switching: 18 autosomal regions (spanning ~3 Mb, ~0.1% of the autosomal genome) differed between NF-LV-CM and F-LV-CM, and 7 regions (~3.2 MB, ~0.1% of the autosomal genome) differed between NF-LA-CM and NF-LV-CM (Supplementary Fig. 11). Importantly, these compartment shifts were not linked to changes in histone modifications or gene expression.

These findings indicate that gene expression differences according to chamber identity and disease are not driven by global chromatin architecture remodeling (TADs, A/B compartments).

We have added this analysis and the supporting figures to the revised manuscript results section (lines 146-157), methods section (lines 841-856) and added Supplementary Fig. 11.

4) Fig. 3d-f: The increased interaction of the S-E with the NPPA promoter in F-LV-CMs is interesting but looks very subtle. Can the authors verify this interaction using a more quantitative method?

Response:

We thank the reviewer for this point and addressed it in two ways in the revised manuscript: we first perturbed the central super-enhancer domain using CRISPRi in hiPSC-aCM to validate the functional relevance of this interaction (Fig. 5f,g).

Second, we fine-mapped the altered interaction pattern of the super-enhancer with the NPPA promoter using a tiled super-enhancer viewpoint analysis (1kb resolution), as shown in the new Fig. 5d. This analysis quantifies the differential interaction of different super-enhancer domains with the NPPA promoter.

We agree that the differences are visually minor at first glance due to the high background level, typically for regions proximal to the viewpoint. In the revised version, we therefore highlight the background levels in the bar plots of our replicate-based HiC analysis, too. Most Hi-C studies performed statistical analysis using Hi-C data merged from multiple biological replicates, which does not account for biological and technical variability. The individual values resulting from the biological replicates displayed in the bar and scatter plots are reproducible, highlighting the reliability and quantitative nature of Hi-C. To our knowledge, this is the first replicate-based analysis of promoter-enhancer interactions, which was made possible by the new version of HiCExplorer with features developed as part of this project. This aligns with recent work comparing Hi-C-based methods to alternative approaches to study chromatin interactions. The quantitative nature of Hi-C data was confirmed by two recent preprints demonstrating highly correlating interaction data between Hi-C and imaging

data from DNA-FISH or fluorophores tagged to interacting loci^{3,4}. However, DNA-FISH showed a lower signal-to-background ratio than HiC³. Therefore, imaging-based methods are not a method of choice for detecting differences between cell states or subtypes, like LA-CM and NF-CM, as studied in this manuscript. Among the different “C-methods”, high-resolution Hi-C is the gold standard for detecting chromatin interactions if the sequencing depth is high and a frequent cutter is used⁵. We used two frequently cutting restriction enzymes with the recognition motifs GATC and GANTC and applied deep sequencing per replicate. We would agree that it is, in general, hard to quantify the interaction strengths of multiple regions interacting with the same promoter, e.g., a distal and a proximal enhancer, due to the bimodal distribution of the signal. Nevertheless, we think Hi-C does allow reliable quantitative detection of interaction strength across samples.

We have updated the results section (lines 247-256), and added Fig. 5.d,f,g.

5) Fig. 5: The authors should provide a table with target genes for the GWAS hits?

Response:

In the revised manuscript, we provide comprehensive lists of PI-CREs (Supplementary Table 7-9) along with PIDs detected for each condition. This allows identification of candidate CREs and target genes by simple intersection with variant coordinates. We opted not to include processed PID-variant intersections as providing such tables for the different conditions and all GWAS data (including sub-threshold variants) studied here would require many large supplementary files, which would make integrating novel variants detected in new GWAS studies hard. Instead, this format provides a more flexible and user-friendly resource for ongoing and future variant annotation efforts.

6) The authors should compare findings to other recent human heart epigenomic and snATAC-seq studies by Zhang et al., PMID: 31427791, Anene-Nzeiu et al. PMID: 32866060, Avantis et al. PMID: 32111823, Foo studies PMID, Hocker et al. PMID: 33990324, Spurrel et al. PMID: 36130500 and Ameen et al. PMID: 36563664.

Response:

We thank the reviewer for this list of relevant manuscripts related to cardiac epigenetic mechanisms.

In the revised version of the manuscript, we integrate the original ATAC data for NF-LV-CM and NF_LA-CM from Hocker et al. (PMID: 33990324). We also noted that, unlike our study, this paper did not report chamber-specific variant assignment to CRE in atrial and ventricular CM, a point we also discussed with co-corresponding author Sebastian Preißl.

The second paper listed from the Ren and Chi labs (with Sebastian Preißl) studied chromatin organization during ES cell to CM differentiation, which is not directly linked to our manuscript centered on adult CM chromatin architecture. As acknowledged in response #7, we plan to do an in-depth analysis of ES-/iPS-derived CM data in comparison to in vivo CM data in a separate study, since this comparison is beyond the scope of this manuscript.

In the course of this revision, we cross-validated the enhancer deletions performed by Anene-Nzeli et al. (PMID: 32866060) using CRISPRi (Supplementary Fig. 14). In an earlier version of this manuscript, we had included cardiac tissue Hi-C data kindly shared by Roger Foo (DOI: 10.21203/rs.3.rs-1897221/v1). Following concerns about potential methodology-

induced biases for comparison and to increase sequencing depth we replaced it with original data from tissue matching the sorted CM samples. Nevertheless, both tissue datasets look identical and show the same differences compared to CM-specific data, highlighting the benefit of cell type-specific analysis and that HiC is reproducible even across different labs and protocols.

The snATAC study of Ameen et al. PMID: 36563664 focuses on the genetics of congenital and developmental processes. This manuscript shows the function of variant enhancers in endothelial cells for the development of CHD and characterizes the regulatory landscape of iPS-CM compared to fetal atrial and ventricular CM. This comparison shows a high correlation of chromatin accessibility between iPS-derived and fetal CM. We discuss this paper in the revised version as part of the introduction and limitation section.

Avantis et al. PMID: 32111823 reports a variant (rs535411) overlapping with an accessible region in iPS-CM. They linked this variant to ACTN2 using ES-CM Hi-C data. We analyzed this locus, and the variant region is accessible in NF-LV-CM, too. However, the interaction with the ACTN2 promoter is not very strong and does not reach our detection cut-offs. Remarkably, a neighboring variant rs1544970 is associated with an ATAC peak in NF-LV-CM and shows a chamber-specific interaction with the ACTN2 promoter in NF-LV-CM, orchestrating the increased ACTN2 expression in NF-LV-CM as compared to NF-LA-CM. We have extended our discussion to highlight variants impacting cardiac gene expression confirmed by enhancer deletion.

We cite Spurrel et al. PMID: 36130500 and Tan et al. Circulation (Foo lab) together with (Dickel et al. Nat. commun.) for the generation of cardiac H3K27ac data of diseased hearts in the introduction. We did not integrate the original data due to the availability of CM-specific data spanning fetal and postnatal maturation and heart failure⁶.

We have done our best to integrate suggested cardiac epigenetic studies related to this work, reflecting on the new comparisons and emphasizing where our findings extend beyond these prior studies.

7) Author should show that hiPSC cardiomyocyte model also display similar epigenetic landscape and chromatin interactions as human ventricular cardiomyocytes to show that hiPSC cardiomyocytes could be used as a model to test studied CREs. This is especially important as the authors have shown that there are significant differences in both epigenomic and chromatin landscape between different types of cardiomyocytes.

Response:

We thank the reviewer for this important point. We agree that hiPSC-CM are not fully mature and do not fully recapitulate the epigenome of adult CM. Given that an in-depth analysis of chromatin and epigenetic conservation across the entire genome is outside of the scope of this manuscript, we have not included this data but plan to address this in a future study. We acknowledge that differences in the epigenetic and chromatin interactions exist, hence a functional analysis in hiPSC-CM or specific subtypes is relevant when the putative CRE-target interaction and the epigenetic states are conserved with adult CM.

Results shown in Reviewer Figure 2 highlight that 76% of PIDs identified in NF-LV-CM were also detected in hiPSC-vCM (a), indicating substantial homology in PID formation. In addition, CpG methylation patterns of PI-CRE detected in NF-LV-CM are highly similar in hiPSC-vCM (b), supporting the conservation of regulatory elements. We further compared

the relative PID interaction strength and the number of PIDs for CM marker genes listed in Supplementary Table 1. This comparative analysis did not reveal significant differences between NF-LV-CM and hiPSC-vCM (c,d).

These findings suggest a high degree of conservation at a large set of functionally relevant loci, supporting the use of hiPSC-CM for CRISPRi-based functional validation. However, we have included a statement in the limitation section acknowledging this point (lines 457-465).

Reviewer figure 1: Comparative analysis of LV-NF-CM and hiPSC-vCM

a) Fraction of NF-LV-CM PIDs detected in hiPSC-vCM

b) CpG methylation heatmap of PI-CRE detected in NF-LV-CM and flanking regions. Plotted are CpG methylation data for NF-LV-CM and hiPSC-vCM.

c,d) Cumulative relative interactions and number of PIDs per CM marker gene (112 genes, PanglaoDB⁷) detected in NF-LV-CM and hiPSC-vCM. Two-tailed Mann-Whitney test.

8) Knockouts of CREs and knock-ins of genetic variants should be performed to validate CRISPRi studies.

Response:

We thank the reviewer for suggesting these control experiments. To validate our CRISPRi silencing methodology, we performed the following experiments and analysis:

1) Cross-validation of published enhancer deletion studies using CRISPRi: We silenced enhancers of *MYL2*, *MYH7*, *SCN5A*, and *KCNH2* using CRISPRi. CRISPRi decreased gene expression of the respective target genes, mimicking the previously reported effects observed upon enhancer excision *in vitro* and *in vivo*⁸⁻¹¹ supporting the robustness of CRISPRi for interrogation of CRE function (Supplementary Fig. 14, Supplementary Fig. 15).

2) Targeted *KCNJ2*-CRE ablation to validate CRISPRi: In this study, we identified a functional *KCNJ2* enhancer (region 4) using CRISPRi. We deleted this region using CRISPR/Cas9-mediated genome editing in hiPSCs. Two independent deletions were generated followed by CM differentiation and both led to a significant downregulation of *KCNJ2* in differentiated hiPSC-vCM, consistent with our CRISPRi results. (Supplementary Fig. 21).

To study the impact of genetic CRE variants on a global scale, we integrate PID data with cardiac eQTL data². This identified 7608 genetic PI-CRE variants affecting gene expression of the interacting gene. As shown in the new Fig. 7h-j, genotype-dependent expression differences were even observed in a chamber-specific manner. For example, rs12028141, a variant overlapping a ventricular PI-CRE, was significantly associated with *NOS1AP*

expression in the left ventricle (P value = $4.14e^{-8}$) but not in the atrial appendage. Likewise, rs12468863 was linked to KCNK3 expression specifically in the atrial appendage (P value = $8.4e^{-7}$) but showed no significant association with KCNK3 expression in the ventricles.

We updated the results sections accordingly (lines 187-193, 344-345, and 322-325), and the methods section (lines 693-713, 729-739, 747-748, and 911-913).

9) Functional electrophysiologic studies should also be performed in hiPSC cardiomyocytes to confirm the impact of perturbing CREs.

Response:

We thank the reviewer for this critical and challenging suggestion which led to novel insights. To directly assess the functional impact of CRE perturbation at the electrophysiological level, we performed automated patch-clamp recordings in hiPSC-vCM following CRISPRi-mediated silencing of the KCNJ2 promoter and two PI-CREs (region 3,4). KCNJ2 encodes the inward-rectifier potassium channel Kir2.1. Silencing of the KCNJ2 promoter led to 90% reduction in IK_1 currents, validating the methodology. Silencing of the distal PI-CREs region 3 and 4 resulted in 39% and 67% reduction in IK_1 current, respectively. These effects closely mirrored the impact of CRISPRi on KCNJ2 mRNA levels. We further checked whether silencing of region 3 and 4 affects the repolarization time using multi-electrode arrays. We observed a prolonged repolarization time compatible with the inhibition of the Kir2.1 currents, highlighting the physiological function of these regions.

These results show for the first time that these distal CREs contribute to functional regulation of ion channel activity (Fig. 8, Supplementary Fig. 20).

These results and the additional CRISPRi experiments performed for the revised manuscript highlight that combining cell-type-specific epigenome and chromatin interaction data with functional perturbation can provide novel insight into the physiological relevance of disease-associated genetic variants.

We updated the results sections accordingly (lines 349-356), and the methods section (756-773).

10) It is somewhat surprising that perturbing AR2 CRE within the SCN5a locus has no functional impact on SCN5a expression when the authors' data and analyses suggest that it should, thus raising concerns about the utility of the authors' findings.

Response:

We appreciate the reviewer's careful attention to this point. Although AR2 is predicted to interact with the SCN5A promoter, its silencing with CRISPRi had no significant impact on SCN5A expression. However, silencing of the neighboring interacting AR1 element did significantly reduce SCN5A gene expression. The Christoffels lab¹¹ reported comparable results for corresponding regions in mice, highlighting a functional evolutionary conservation of these enhancer domains.

We interpret the lack of efficient silencing via AR2 not as a contradiction to our approach but rather as an illustration of the context-dependent and non-binary nature of CRE function. AR2 activity may be redundant with nearby regulatory elements, developmental stage-specific, or active in specific conditions. It has been reported that not all CREs are detected

to be functionally active under baseline conditions for several biological systems. Chromatin interaction data and epigenetic data are essential to prioritize active CRE candidates. During the revision, we silenced 9 additional CREs, and only one did not significantly reduce target gene expression. This high fraction of positive read-outs with this approach exceeds, in our experience, the outcome of studies targeting CREs without prior knowledge of epigenetic and chromatin interaction data. In the case of these loci, CRISPRi had no detectable impact on chromatin organization (Supplementary Fig. 14), Since it can be applied to matured hiPSC-CM, our CRISPRi strategy also avoids interfering with the cell differentiation process.

Minor

1) Better figure labeling: e.g. for some figure panels one has to look at the figure caption to know what the different colors mean.

Response:

We thank the reviewer for the critical note. We have updated all figures to include the color legends.

2) Line 136-139: Define PID better (how is this different from a loop?). PIDs span 54 MB and overlap with 30k CREs spanning 24 MB. What is the remaining 30 MB if not CREs?

Response:

Our study defines promoter-interacting domains (PIDs) as continuous genomic intervals exhibiting statistically significant interaction with a given gene promoter based on virtual viewpoint analysis. These domains are incrementally increased during the analysis, starting from an individual bin. This analysis is performed independently from the annotation of regulatory elements. Therefore, PIDs do not necessarily overlap with CRE elements. In reality, most overlap with individual or multiple CREs, interspaced, e.g., by non-regulatory chromatin. This allows the detection of chromatin interactions independent of epigenetic marking. We have integrated additional plots into Supplementary Fig. 13 visualizing the chromatin state of PIDs, and CREs in more detail to address your question.

This approach differs from e.g. the ABC model, which integrates different data modalities and reports loops based on H3K27ac marking of individual ATAC peaks.

In contrast to PIDs, loops are discrete point-to-point contacts and reflect mostly interacting TAD boundaries.

3) Lines 161-165: This section is difficult to follow. It's not clear how can you have diminished CTCF-mediated chromatin interactions without impacting higher-order chromatin organization? Are these CTCF sites not at TAD boundaries?

Response:

We agree that we did not explain the related supplementary figure in detail. In the revised version, we extended this section in the main manuscript to improve clarity. Indeed, only 8% of differential PIDs (233) overlap TAD boundaries. This may explain the lack of dynamic TADs. So far, we cannot explain the relevance of the CTCF-positive PIDs in failing CM, given that they are unrelated to gene expression changes. Further studies are needed to clarify the underlying mechanisms. We therefore report this observation only as part of Supplementary Fig. 18.

4) Supplementary Fig 11: There's a clear difference for NF-LA LMRs, but very marginal for NF-LV and F-LV. Can the authors do statistics for these comparisons? Furthermore, it could be interesting to plot ChIP-data, in particular H3K27ac, for the LMRs, as there may be differences in enhancer activity that are not captured by CpG methylation.

Response:

We changed the analysis to a replicate-based method (Metilene¹²). These results confirm that CpG methylation of CREs is minimally affected in failing CM as we have already reported for LMRs and additional genomic elements using an alternative annotation strategy⁶. In contrast, CpG methylation of CREs in LA and LV CM shows substantial differences. We added this analysis to the revised manuscript (Supplementary Fig. 12).

As suggested, we now also include an analysis of differential H3K27ac marking of CREs in failing CM and link these changes to differential gene expression using our chromatin interaction data (Fig. 6c-d). We confirm the functional association for a dynamically 27ac-marked HEY2 CRE using CRISPRi (Fig. 6e and Fig. 4c).

We updated the results sections accordingly (lines 273-282), and the methods section (lines 799-801, and 819-820).

Reviewer #2 (Remarks to the Author):

In this MS, the authors employ Hi-C to analyse chromatin interactions in healthy atrial and ventricular cardiomyocytes as well as failing ventricular CM. By also performing this analysis in ventricular tissue – which comprises multiple cell types, they illustrate that analysis specifically of the CM compartment is required to gain insight into how chromatin organisation can relate to other epigenetic marks and transcription in CM. This requirement to specifically analyse CM and not tissue builds on their elegant studies on cardiac epigenetics. In this current MS, the authors demonstrate chamber specific chromatin interactions that were related to associated transcription and transcription factor binding. Analysis of locations of SNP associated with cardiac disease determined these to be localised to chamber specific chromatin interacting regions. However, these did not appear to change with disease. Interestingly CRISPRi disruption of sites of interaction with the KCNJ1 locus that coincided with identified SNPs, decreased KCNJ1 expression in iPSC-CM suggesting functional importance. This MS provides important new information regarding the regulation of gene expression across the chambers of the heart and how it may be affected by SNPs located at cis regulatory elements. Only through correlating these chromatin interactions specifically within CM and establishing the functional role of the CREs identified could these findings be reached.

Response:

We thank the reviewer for the time and effort spent reviewing the paper. We appreciate the acknowledgment of our approach and of the insights gained from integrating CM-specific epigenetic, chromatin structure, and gene expression profiles

As you can see from the revised manuscript and the responses to the other reviewers, we added substantial new data supporting our findings and providing novel functional and mechanistic insights, as summarized on page 1 of this document.

There are some questions which require addressing:

1- In Fig. 2d, only two groups are being compared, but the Kruskal-Wallis test was used. Since the Kruskal-Wallis test is generally used for comparing more than two groups, shouldn't the Wilcoxon rank-sum test be used instead?

Response:

We thank the reviewer for highlighting this statistical question.

Both tests can be used to compare two groups. In fact, both tests gave the same statistical confidence level in Fig. 2d. Given that the values of Fig. 2d are discrete, we considered the Kruskal-Wallis test to be more appropriate. After statistical consultation, we realized that both tests can be used for Fig. 2b-d. In the revised version, we use the Kruskal-Wallis test for all panels to avoid confusion. We have updated the figure and legend accordingly.

2- In Fig.3, the H3K27ac-marked super-enhancer is located upstream of both NPPA and NPPB, but the focus was primarily on NPPA, despite the RNA expression data showing that NPPB was also affected. Was this due to interactions of CRE only with the NPPA promoter? According to the Human cell atlas, both NPPA and NPPB are expressed in atrial cardiomyocytes, but NPPB is also expressed in some ventricular CM. Wouldn't it be valuable to discuss NPPB in more detail in the context of ventricular CM as well? It seems that the RNA peaks for NPPA and NPPB don not fully align with the gene locus. Perhaps the figure needs adjusting.

Response:

We thank the reviewer for noticing the gene track alignment error, which we have corrected in the revised figure.

We have also added NPPB viewpoints for all groups to Fig. 5 as well as statistical details for the dynamic interaction domains (Fig. 5e). This analysis reveals that, specifically in atrial CM the super-enhancer shows strong interaction with the NPPB promoter. Remarkably, this interaction peaks in the central part of the super-enhancer, while the interaction with the NPPA promoter spans a larger domain in atrial CM. These results show that the interaction pattern with the super-enhancer is distinct for both genes. Furthermore, the central super-enhancer domain shows increased NPPA and NPPB promoter interactions in F-LV-CM, too. We therefore added functional validation data for the central super-enhancer region using CRISPRi in hiPSC-derived atrial CM. Silencing of this region resulted in an 82% reduction in NPPB expression, confirming its enhancer activity. Interestingly, we also observed a 31% reduction in NPPA expression, suggesting that this element modulates the transcription of both genes. From our point of view, these functional data are in good agreement with the chromatin interaction data and support that different segments of the super-enhancer are engaged in a chamber- and disease-specific manner, allowing for dynamic regulation of both NPPA and NPPB.

We incorporated these findings into the revised results section (lines 247-256) and updated the corresponding Figures.

3- In Fig. 5, analysis was restricted to promoter interacting CREs in non-failing atrial and ventricular CM. This analysis identified enrichment for disease associated SNPs in these promoter-interacting CREs that were related to disease in the related chamber. Would a stronger relationship with disease related SNPs be observed in PI-CRE in failing CMs.

Response:

Considering that disease-associated genetic variants are enriched in HF-associated CREs implies that the genetic burden is modified by the onset of heart disease. To address this, we performed a GWAS analysis of HF-associated CREs and did not find a clear enrichment of genetic variants for any tested genetic traits compared to NF-LV-CM CREs. This indicates that HF-associated CREs do not seem to modulate genetic disease risk factors in human LV-CM. In contrast, a strong enrichment was observed for QT-associated variants specifically in NF-LV-CM. Therefore, we focused on these genetic risk factors in our manuscript. The results of our GWAS analysis are summarized in Reviewer Fig. 1, which is attached below.

Reviewer Fig. 1: Heatmap displaying the significance of the enrichment calculated for genome-wide association study (GWAS) variants compared to random variants for NF-LV-CM PI-CREs and differential PI-CREs in F-LV-CM as compared to NF-LV-CM. Shown are results for various organ disease traits from the NHGRI-EBI GWAS catalog.

4- Similarly, it would be informative and add weight to the relevance of the interaction, if RNA levels, ATAC peaks, H3K27ac peaks were provided for KCNJ2 for failing as well as non-failing LV CM samples in Fig. 6a.

Response:

We thank the reviewer for this suggestion and have integrated a new Supplementary Fig. 19 in the revised manuscript to display RNA, CpG methylation, H3K27ac, and chromatin interactions. The ATAC-seq data presented for NF-LA-CM and NF-LV-CM is pseudobulk data extracted from snATAC-seq data published by Hocker et al.¹⁰, which did not cover failing LV. We also checked additional canonical histone marks (H3K4me3, H3K27me3, H3K9ac, H3K4me1, H3K27me3, H3K9me3, H3K36me3) from a previous study⁶ and did not identify relevant differences in chromatin signatures. Importantly, promoter-interacting domains detected in NF-LV-CM were also detectable in F-LV-CM (Supplementary Fig. 19)

5- In Fig 6b, the silencing of KCNJ2 promoter-interacting elements 3 and 4 significantly reduced KCNJ2 gene expression, yet Fig 6e shows no significant effect on the interaction frequency between these regions and the KCNJ2 promoter. Supplementary Fig 4 suggests there are only four differentially promoter-interacting domains (PIDs) for KCNJ2, possibly including elements 3 and 4. How can the silencing of interacting elements 3 and 4 reduce gene expression without affecting the promoter interaction frequency? What might explain this apparent disconnect between gene expression and promoter interaction?

Response:

We thank the reviewer for raising this important question.

CRISPRi silences regulatory elements by inducing inaccessible chromatin, as shown in Fig. 8d and in a previous publication from our group¹³. Consequently, these elements are inaccessible for DNA-binding proteins, including transcription factors. To our knowledge, it has not been demonstrated whether CRISPRi-mediated silencing of CRE impacts the interaction with interacting promoters. We therefore added experimental data to address this point.

Initially, we observed that the CRISPRi of two KCNJ2 promoter-interacting CREs has no major impact on PID contact frequencies (Supplementary Fig. 14f). To generalize this finding, we target two additional CREs using CRISPRi in the revised manuscript. These CREs interact with the promoters of KCNH2, and TBX5. CRE silencing resulted in transcriptional repression of the respective target genes (Supplementary Fig. 14a, Fig. 4d). However, the PID contact frequencies were not affected (supplementary Fig. 14d,e). These experiments show that physical chromatin interactions of promoter-CRE interactions are not affected by CRISPRi. A comparable observation was recently published for CRISPRa, where activation of inactive CRE did not affect the 3D genome organization¹⁴. This observation underscores that CRE's physical chromatin interaction and activity are functionally separable.

This supports a model in which enhancer-promoter looping may be permissive or pre-established but insufficient for transcriptional activation unless the cis-regulatory element is epigenetically active.

We updated the results sections accordingly (lines 197-202).

6- Low I_{K1} is a feature of iPS-CM contributing to their immature AP. Given that KCNJ1 is the contributing channel to this current, would its expression not also be basally very low in iPS-CM. The authors show relative change in expression of KCNJ1 following CRISPRi. The relevance of these changes may however not be functionally important if expression is very low and may also overestimate the effect of CRISPR – i.e. CRISPR does not need to be very effective to reduce KCNJ1 expression. Could the levels of mRNA expression be shown and how do these compare to levels in adult vCM.

Response:

We appreciate the reviewer's thoughtful point. We assessed KCNJ2 gene expression in hiPSC-vCM relative to hiPSC-aCM and adult ventricular and atrial tissue to address this concern. We found that KCNJ2 transcript levels are substantially higher in hiPSC-vCM than in hiPSC-aCM. However, levels of hiPSC-vCM did not fully reach adult ventricular tissue, indicating a not fully mature transcriptional profile of hiPSC-vCM (Supplementary Fig. 20a). To confirm the functional levels of KCNJ2 in hiPSC-vCM, we performed high-throughput automated patch-clamp analysis, which showed that 72.5% of hiPSC-vCM exhibited detectable I_{K1} currents attributable to Kir2.1 (Supplementary Fig. 20b-e), aligning with our previous studies¹⁵. This demonstrates that our hiPSC-vCM expressed functional KCNJ2 levels.

To directly assess the functional impact of CRE-silencing on Kir2.1 currents, we performed automated patch-clamp recordings in hiPSC-vCM following CRISPRi-mediated silencing of the KCNJ2 promoter and two PI-CREs (region 3 and 4). Silencing of the KCNJ2 promoter led to a 90% reduction in I_{K1} currents, validating our methodological strategy. Silencing of the distal PI-CREs region 3 and 4 resulted in 39% and 67% reduction in I_{K1} current, respectively. These effects closely mirrored the impact of CRISPRi on KCNJ2 mRNA levels. We further checked whether silencing of region 3 and 4 affects the repolarization time using multi-electrode arrays. We observed a prolonged repolarization time compatible with the inhibition of the Kir2.1 currents, highlighting the physiological function of these regions. These results show for the first time that these distal CREs contribute to regulating ion channel activity and repolarization time (Fig. 8e-f, Supplementary Fig. 20f-g).

These results and the additional CRISPRi experiments (Fig. 4c,d, 5f,g, Supplementary Fig. 14a-c) highlight that combining cell-type-specific epigenome and chromatin interaction data with functional perturbation uncovers the physiological relevance of disease-associated non-coding regulatory elements.

We updated the results sections accordingly (lines 333-335, and 349-356), and the methods section (lines 756-773).

Reviewer #3 (Remarks to the Author):

In “Linking Chamber-Specific Spatial Chromatin Interactions to Disease Variants and Gene Programs in Human Cardiomyocytes” Haydar et al. conduct genome wide chromatin interaction mapping in ventricular vs atrial cardiomyocytes and ventricular cardiomyocytes from failing left ventricles. They then conducted multiple analyses related to cis regulatory element – promoter interactions and discovered chamber specific interaction that correlated with specific gene expression. Moreover, chamber specific interactions were more likely to be found in regions of disease associated genetic variants – e.g. AFIB associated variants linked with cis regulatory elements enriched in the atria vs ventricle. Additional CRISPRi experiments validated the importance of these promoter-cis regulatory element interaction on target gene expression. While this is not the first manuscript to associate genetic variants with cis regulatory elements, it is the first to show chamber specific cis regulatory elements and links to variants of chamber specific disease etiology. This work will impact workflows for investigating/validating genetic variants. These points are sufficiently impactful to indicate this will be a worthwhile addition to the literature and of appropriate scope for consideration by Nature Communications.

Response:

We thank the reviewer for the encouraging comments and suggestions to improve the quality and impact of this study. As you can see from the revised manuscript and the responses to the other reviewers, we added substantial new data supporting our findings and providing novel functional and mechanistic insights, as summarized on page 1 of this document.

The biggest surprises in the data for this reviewer were that 1) largely the “...interactions were not affected in CMs of failing hearts and not linked to pathological gene expression on a genome wide scale.” And 2) CRISPRi of CREs failed to disrupt interactions despite impact on gene expression and chromatin accessibility (ATAC).

These findings could have profound implications for current models/understanding of gene expression regulation in the heart. Are the authors confident that the techniques used and data generated were sufficiently robust to draw these conclusions (positive control for 6e would help)? If so, please provide direct language in rebuttal explaining why and direct language in manuscript highlighting the implication of the findings.

Response:

1) We data confidently shows that rewiring in failing CM is restricted to a few genes (including NPPA) and not generally linked to pathological gene expression. Unpublished data from our lab indicate this is not the case for other cardiac cell types, in which significant dynamic rewiring events can be associated with pathological gene expression. The same holds for the chamber-specific chromatin organization of CM reported here. These findings support the fact that our methodology can capture such events. However, we agree that the blunt absence of an effect is statistically hard to prove due to a lack of statistical methods. The revised manuscript provides evidence that differential H3K27ac marking of CRE elements is linked to differential gene expression via preformed chromatin interactions. We tested two disease-associated CREs using CRISPRi and could confirm their regulatory function.

2) CRISPRi silences regulatory elements by inducing inaccessible chromatin, as shown in Fig. 8d and in a previous publication of our group¹³. Consequently, these elements are inaccessible for DNA-binding proteins, including transcription factors. To our knowledge, it

has not been demonstrated whether CRISPRi-mediated silencing of CRE impacts the interaction with interacting promoters.

Initially, we observed that CRISPRi-mediated silencing of two KCNJ2 promoter-interacting CRE has no significant impact on PID contact frequencies (Supplementary Fig. 14f). We target two additional CREs using CRISPRi in the revised manuscript to generalize this finding. These CREs interact with the promoters of KCNH2 and TBX5. CRE silencing resulted in transcriptional repression of the respective target genes. However, the PID contact frequencies were unaffected (Supplementary Fig. 14a,d,e, Fig. 4d). These experiments show that physical chromatin interactions of promoter-CRE interactions are not affected by CRISPRi. A comparable observation was recently published for CRISPRa, where activation of inactive CRE did not affect the 3D genome organization, which was measured using a high-resolution tiling DNA-FISH probe approach (ORCA)¹⁴. This observation underscores that CRE's physical chromatin interaction and activity are functionally separable.

This supports a model in which enhancer-promoter looping may be permissive or pre-established but insufficient for transcriptional activation unless the enhancer is epigenetically active.

We updated the results sections (lines 197-202, 254-256, and 273-282), the methods section (lines 819-820), and clarified this interpretation in the discussion section (lines 414-419), accordingly.

Minor issues indicated below:

“Histone code” bar in multiple figures appears inaccurate in terms of RNA production.

Response:

We replotted the figures in the revised manuscript. The NPPA locus figure had a misaligned gene track. We added the histone code legend to all figures. The histone code annotation was generated using ChromHMM¹⁶ and is visualized in Supplementary Figure 2a.

Sentence from line 271-273 missing at least one word.

Response:

We edited the paragraph in the revised version of the manuscript.

What is the interpretation of interaction domain size difference between CM and Tissue 2C

Response:

We agree that interpreting the interaction domain size is not trivial since it represents the cumulative PID size for each assessed gene. For example, for the MFN2 gene (Supplementary Fig. 5), we detect only two PIDs in LV-tissue and six PIDs in the NF-LV-CM. In addition, PIDs in NF-LV-CM were frequently broader, covering multiple clustered CREs, including super-enhancers. Identifying broad interactions spanning clustered CREs does not work well in tissue data, likely because the capture of cell-type specific interaction information is diluted by the mix cell types in contrast to shared interactions.

Please articulate potential limitations of comparing current HiC to epigenetics data from another study.

Response:

Potential biases resulting from integrating and comparing Hi-C data with epigenetic data from other sources are different bioinformatic or wet lab techniques, e.g., approaches used for cell isolation or sorting, and most importantly, data quality. We therefore aimed here to integrate data generated using comparable methods. This holds especially true for CM-nuclei sorting. All epigenetic data from sorted nuclei were generated by our group either for this study or for previous studies⁶. We also used unified bioinformatic pipelines for all data sets. Only the integrated NF-LA-CM and NF-LV-CM ATAC data were obtained from a snATAC-seq study¹⁰. However, these data were of high quality and in good agreement with previously published ATAC data from FACS-sorted nuclei generated by the Martin lab¹⁷, highlighting the importance of evaluating the technical quality of each internal and external data set. Another data integration problem is that the data were frequently obtained from different biological samples. To reduce this cofounder, we used, in several cases, material from the same donor for the different data sets.

We have included a statement in the limitation section acknowledging this point (lines 466-470).

“A recent study shows that enhancer promoter interactions are becoming more instructive during muscle terminal differentiation.” Instructive to who/what?

Response:

We agree that the term "instructive" is not clearly explained. The cited manuscript emphasized that active enhancers gain functional relevance during maturation due to increased chromatin interactions and activity. We removed this sentence and the citation from the revised manuscript.

Labeling in some figures could be improved for interpretability – Suggest adding color code for groups to individual figure panels where appropriate.

Response:

We have updated all figures to include color legends.

While the methods section is generally excellent, this reviewer has some confusion regarding the GWAS and variant vs proxy variant identification using a 1Mb window.

For the proxy variants found at extreme distances shown to overlap CRE, was that CRE associated with a promoter nearer the lead variant? Comment may be outside of scope.

Response:

For this manuscript, we independently used the lead and proxy variants (LD $r^2 > 0.7$, max. distance 1 Mb) and thus cannot fully address the comment based on the presented data.

We agree that it would be of great interest to study whether lead and proxy variants interact generally with the same target promoter or whether they interact with distinct or shared promoters. A multi-contact analysis of GWAS hits would therefore be of great interest, but it is indeed beyond the scope of this project.

Interestingly, both lead and proxy variants located in PIDs show a unimodal distribution of distances to gene promoters (Reviewer Fig. 3a) and follow the distribution of PID distance relative to their target promoter (Supplementary Fig. 13d). This supports the use of chromatin interaction data to decode the regulatory variants located at far-away distances from the target genes.

Among 650 genes interacting with QT variant-containing PID, 62% interact with both a lead variant and one or more of its proxies (Reviewer Fig. 3b). This supports a hypothesis that genes often interact with multiple variants within a functional LD block, rather than being influenced by a single GWAS-identified lead SNP. This indicates that regulatory effects are broadly distributed across physically linked variants, reinforcing the importance of incorporating chromatin structure in interpreting GWAS.

Reviewer Fig. 3: Variant-gene assignment using chromatin interaction data supports both lead and proxy variants.

a Distribution of genomic distances of lead (pink) and proxy (white) QT-associated variants¹⁸ to their target gene promoters.

b Categorization of genes interacting with QT-associated lead and proxy variants¹⁸.

As indicated introduction contains a sentence indicating “interactions were not affected in CMs of failing hearts and not linked to pathological gene expression on a genome wide scale” yet discussion contains sentence indicating “Our data suggests that CTCF-mediated chromatin interactions are affected in terminal heart failure in CM” then goes on to discuss NPPA regulation. The “our data...” sentence should be qualified in some way to indicate lack of support for this observation in the genome-wide analysis.

Response:

We thank the reviewer for highlighting this discrepancy. We agree that the sentence in the introduction is misleading. We updated it in the revised manuscript to clarify that we observe rewiring-associated gene expression changes only for 8 genes and that further events were unrelated to gene expression changes but to some degree linked to CTCF sites.

References:

- 1 Kosicki, M. *et al.* VISTA Enhancer browser: an updated database of tissue-specific developmental enhancers. *Nucleic Acids Res* (2024). <https://doi.org/10.1093/nar/gkae940>
- 2 Consortium, G. T. The GTEx Consortium atlas of genetic regulatory effects across human tissues. *Science* **369**, 1318-1330 (2020). <https://doi.org/10.1126/science.aaz1776>
- 3 Li, Y., Zou, F. & Bai, L. Hi-C Calibration by Chemically Induced Chromosomal Interactions. *bioRxiv* (2024). <https://doi.org/10.1101/2024.12.09.627644>
- 4 Jusuf, J. M. *et al.* Genome-wide absolute quantification of chromatin looping. *bioRxiv* (2025). <https://doi.org/10.1101/2025.01.13.632736>
- 5 Consortium, D. N. *et al.* An integrated view of the structure and function of the human 4D nucleome. *bioRxiv* (2024). <https://doi.org/10.1101/2024.09.17.613111>
- 6 Gilsbach, R. *et al.* Distinct epigenetic programs regulate cardiac myocyte development and disease in the human heart in vivo. *Nat Commun* **9**, 391 (2018). <https://doi.org/10.1038/s41467-017-02762-z>
- 7 Franzen, O., Gan, L. M. & Bjorkegren, J. L. M. PanglaoDB: a web server for exploration of mouse and human single-cell RNA sequencing data. *Database (Oxford)* **2019** (2019). <https://doi.org/10.1093/database/baz046>
- 8 Dickel, D. E. *et al.* Genome-wide compendium and functional assessment of in vivo heart enhancers. *Nat Commun* **7**, 12923 (2016). <https://doi.org/10.1038/ncomms12923>
- 9 Anene-Nzelu, C. G. *et al.* Assigning Distal Genomic Enhancers to Cardiac Disease-Causing Genes. *Circulation* **142**, 910-912 (2020). <https://doi.org/10.1161/CIRCULATIONAHA.120.046040>
- 10 Hocker, J. D. *et al.* Cardiac cell type-specific gene regulatory programs and disease risk association. *Sci Adv* **7** (2021). <https://doi.org/10.1126/sciadv.abf1444>
- 11 Man, J. C. K. *et al.* An enhancer cluster controls gene activity and topology of the SCN5A-SCN10A locus in vivo. *Nat Commun* **10**, 4943 (2019). <https://doi.org/10.1038/s41467-019-12856-5>
- 12 Juhling, F. *et al.* metilene: fast and sensitive calling of differentially methylated regions from bisulfite sequencing data. *Genome Res* **26**, 256-262 (2016). <https://doi.org/10.1101/gr.196394.115>
- 13 Laurette, P. *et al.* In Vivo Silencing of Regulatory Elements Using a Single AAV-CRISPRi Vector. *Circ Res* **134**, 223-225 (2024). <https://doi.org/10.1161/CIRCRESAHA.123.323854>
- 14 Jensen, C. L. *et al.* Long-range regulation of transcription scales with genomic distance in a gene-specific manner. *Mol Cell* **85**, 347-361 e347 (2025). <https://doi.org/10.1016/j.molcel.2024.10.021>
- 15 Seibertz, F. *et al.* Electrophysiological and calcium-handling development during long-term culture of human-induced pluripotent stem cell-derived cardiomyocytes. *Basic Res Cardiol* **118**, 14 (2023). <https://doi.org/10.1007/s00395-022-00973-0>
- 16 Ernst, J. & Kellis, M. Chromatin-state discovery and genome annotation with ChromHMM. *Nat Protoc* **12**, 2478-2492 (2017). <https://doi.org/10.1038/nprot.2017.124>
- 17 Zhang, M. *et al.* Long-range Pitx2c enhancer-promoter interactions prevent predisposition to atrial fibrillation. *Proc Natl Acad Sci U S A* **116**, 22692-22698 (2019). <https://doi.org/10.1073/pnas.1907418116>

REVIEWER COMMENTS

Reviewer #2 (Remarks to the Author):

In response to the reviewers' comments, the authors have responded with a substantial revision of their MS including new data.

I am generally satisfied with the responses to my questions.

Answer: Thanks for the positive feedback on the revision of our manuscript and the confirmation that our additional data addressed your questions.

One remaining issue – I, as well as other reviewers comment on the choice of KCNJ2 (and its enhancers) as a target that is followed up upon with functional assays. This is followed up on as notably, the authors do not show any HF related interactions – perhaps owing to insufficient sample number.

Answer: Indeed, we did not observe significant enhancer rewiring at the KCNJ2 locus in HF samples (Supplementary Fig. 20). For this analysis, we included n = 8-9 samples, which provides robust statistical power. While we emphasize the importance of annotated enhancers due to their regulatory significance and enrichment in LQT-associated variants, we cannot rule out the possibility that rewiring of this locus is disease-relevant in a subset of HF cases. Testing this hypothesis would require substantially larger sample cohorts, as we now state in the revised limitations section.

Notably, experiments were performed to knockdown expression of a transcription factors and NPPA but consequences for hypertrophic remodelling was not performed.

Answer: Thanks for highlighting the results added during the revision. We agree that perturbing enhancer elements holds the potential to modulate hypertrophic remodeling and HF phenotypes. We tested different hiPSC-CM hypertrophy models but have not yet identified an in vitro system that allows for HF-related phenotypic assessment following enhancer perturbation. Given these limitations of current human cellular models of HF, this aspect lies beyond the scope of this study.

Given this less related analysis of KCNJ2 to HF and link to LQT rather than HF related genes, I would modify the Intro and discussion.

Sudden cardiac death is commonly associated with HF as are other arrhythmic disease, which cardiomyocytes may be predisposed to under conditions of altered KCNJ2. This link to arrhythmia should be made rather than solely focus on HF.

But – this arrhythmia may also be a consequence of hypertrophic remodeling. It may also be argued that a longer APD could signal induction of a hypertrophic response. This could be discussed.

Answer: We agree, HF and LQT are linked risk factors of arrhythmia and represent mutual risk factors resulting in an overlapping syndrome including a predisposition for sudden cardiac death. We discuss this interplay in the revised discussion. However, the genetic risk factor signals of HF- and Arrhythmia-related traits are distinct. Consistent with this, we didn't observe chromatin rewiring at QT-associated loci in HF, as analyzed in response to points raised by the reviewers in the first revision round and visualized for the KCNJ2 locus in Supplementary Fig.20. Testing the impact of LQT on chromatin organization is of high interest, but is currently beyond the scope of this manuscript due to the inaccessibility of cardiac tissue from patients with an LQT syndrome.

In the initial version of the manuscript, HF was only part of the supplementary material (despite one plot related to NPPA). We extended the HF-related analysis and incorporated it into the main manuscript following requests from all three reviewers.

In line with your advice, we modified the discussion and included the limitations regarding the HF-associated rewiring in the corresponding section. Furthermore, in the current version of the abstract, we have limited the mention of HF-related analysis to one sentence.

Reviewer #3 (Remarks to the Author):

Authors have addressed my concerns. Will be a solid addition to the literature.

Answer: Thank you for the positive feedback, reviewing the manuscript and recommending publication.

Reviewer #4 (Remarks to the Author):

This study employs advanced multi-omics techniques (Hi-C, Methyl-seq, ChIP-Seq, ATAC-seq) to comprehensively investigate the epigenomics and 3D chromatin architecture of healthy human atrial and ventricular cardiomyocytes (CMs) as well as failing ventricular CMs. The authors functionally validated promoter-interacting CREs of the MYL2, MYH7, KCNH2, and SCN5A, and prototypic KCNJ2 genes in hiPSC-CMs using CRISPR interference (CRISPRi) approaches. These multi-omics efforts reflect tremendous work, and the additional data provided to address previous reviewer comments have strengthened the manuscript. However, several key issues still need to be addressed before publication.

Answer: Thank you for the positive feedback and for reviewing the manuscript.

Major

1. Unclear central goal: The current manuscript doesn't clearly articulate the central goal, hypothesis, or main research question. This lack of clarity makes it difficult to follow the purpose of the individual sub-sections in the Results. Please revise the abstract, Introduction and Results sections to clearly define the study's overarching objective.

Answer: Thank you for this suggestion. We revised the indicated chapters as suggested to clearly state the overarching study aim and the objectives, as well as the conclusions.

2. Value added by chamber-specific Hi-C: From the title, it appears the authors intend to demonstrate that chamber-specific chromatin architecture revealed by Hi-C provides unique insights for functional interpreting disease associated Cis-regulatory elements. If this is correct, it would be helpful to explicitly summarize key discoveries that were uniquely revealed by chromatin Hi-C sequencing, but not by other epigenetic or transcriptomic profiling.

Answer: Previous studies using epigenetic analysis of heart tissue or cardiomyocytes did not show a difference in GWAS signals in atrial versus ventricular specimens, even in a snATAC-seq study of human atrial and ventricular tissue¹. Our integration of chromatin interaction and epigenetic data fills this gap by demonstrating that chamber-specific interactions not only explain differential gene expression but also uncover a distinct enrichment of GWAS signals in promoter-interacting regulatory elements of atrial and ventricular CM. This distinct enrichment of GWAS variants is causally linked to the disease mechanism.

Importantly, Hi-C data were essential to link disease-associated CREs to their distal target genes, since more than 77% of CRE-Promoter interactions span single or multiple genes, and this fraction rises to 86% for CREs harboring sub-threshold QT-associated variants. We added these findings to the revised manuscript (Supplementary Fig. 13e), and expanded the discussion to emphasize the unique added value of Hi-C.

New Supplementary Fig.13e: Characterization of PIDs, PI-CREs and PN-CREs.

e Percentage CRE-promoter interactions spanning reference genes. Data is shown for all detected PI-CREs and those associated with QT-associated genetic variants of CREs according to Nauffal et al.², applying sub-threshold statistical criteria.

3. Hi-C resolution concern: On page 5 line 151, the manuscript states that “principal component analysis of the Hi-C contact matrices revealed minimal differences in chromatin compartmentalization among groups”, and the author conclude that “both TAD structure and A/B compartmentalization are highly conserved across CM subtypes despite associated transcriptional and epigenetic changes” (line 157). However, considering that a 1 kb bin size was used for Hi-C matrices preprocessing, but larger bin sizes (50kb for A/B compartment and 100kb for TAD) were used for downstream analyses, this raises the possibility that the chosen resolutions may not be sufficient to detect chromatin architecture changes in diseased hearts. Please justify the chosen resolutions and discuss their limitations.

Answer: Thanks for the comment. Topologically-associating domains (TADs) are large genomic domains, typically between 500 kb–1 Mb³, and therefore do not require very high resolution for robust identification. For the initial analysis, we used 100kb bin matrices. Following your comment, we repeated this analysis using smaller bin sizes (10-50 kb) and did not identify alterations in TAD structures, supporting our initial analysis. Notably, we used HiCExplorer⁴ for the identification of TADs. HiCExplorer utilizes windows of varying bin sizes to ensure reliable identification of TADs. We updated the related figures and plot data obtained using 50kb bin sizes. We further confirmed the results visually using a range of bin sizes, since differential TAD analysis is challenging if the expected differences are minor compared to those between cell types. Using this approach, we confirmed the absence of alterations in TAD structures between groups and samples.

A/B compartments span individual or multiple TADs. For the identification of A/B compartments, we used HOMER⁵ using the default resolution of 50kb. This resolution is widely established for identifying these multi-megabase chromatin structures³. Lowering the resolution exponentially increases the computing time without providing additional insights due to the large genomic distances of A/B compartments. The selected bin sizes are thus suitable for obtaining reliable information about these higher-order chromatin structures.

Furthermore, based on our experience, a TAD and A/B compartment analysis using the tools employed in this study with Hi-C data at high resolution (1-5 kb bins) yields

unintentional identification of rewiring events/switches to the level of individual loops, including CRE-promoter interactions, which supports the chosen larger bin sizes. As noted in the revised limitations, new Hi-C analysis methods may reveal additional chromatin organization layers and features in future work. The deep data presented in this study would provide a valuable resource for such studies too. We updated the Supplementary Fig. 11 and the corresponding sections in the main manuscript accordingly.

Updated Supplementary Fig.11e,f: Higher-order chromatin structure of bulk cardiac nuclei, NF-LV-CM, NF-LA-CM, and F-LV-CM.

e Heatmap of Pearson correlation coefficients of TAD separation scores.

f Average plots and heatmaps of TAD separation scores are shown for TAD boundaries of NF-LV-CM across the different CM subtypes.

4. Suitability of hiPSC-CMs for functional validation: hiPSC-CMs were used for functional validation experiments. Since the authors acknowledge the maturation limitations of hiPSC-CMs, it would strengthen the study to compare the 3D chromatin architecture of primary CMs and hiPSC-CMs, particular at the loci studied, to assess whether hiPSC-CMs are a suitable model for those chromatin architecture functional assays.

Answer: We thank the reviewer for this important point. We agree that hiPSC-CM are not fully mature and do not fully recapitulate the epigenome of adult CM. Given that an in-depth analysis of chromatin and epigenetic conservation across the entire genomic landscape is outside of the scope of this manuscript, we have not included these data, but plan to address this in a future study.

In Reviewer Figure 1,2, we represent the promoter interactions of all the perturbed loci shown in Fig. 4c,d (HEY2, TBX5), Fig. 5f,g (NPPA, NPPB) and Fig.8b (KCNJ2) in hiPSC-vCM or hiPSC-aCM. These confirm that the interactions of the studied regions are conserved between primary and hiPSC-derived CM.

Reviewer Figure 3 highlights that 76% of PIDs identified in NF-LV-CM were detected in hiPSC-vCM (a) on a genome-wide scale, indicating substantial homology in PID formation. In addition, CpG methylation patterns of PI-CRE detected in NF-LV-CM are highly similar in hiPSC-vCM (b), supporting the conservation of regulatory elements. We further compared the relative PID interaction strength and the number of PIDs for CM marker genes listed in Supplementary Table 1. This comparative analysis did not reveal significant differences between NF-LV-CM and hiPSC-vCM (c,d). These findings suggest a high degree of conservation at a large set of functionally relevant loci, supporting the use of hiPSC-CM for CRISPRi-based functional validation studies.

Therefore, we have included a statement acknowledging the immaturity of hiPSC-CMs as a limitation.

Reviewer figure 1: Chromatin interaction analysis of perturbed domains in hiPSC-vCM

a,b Shown are gene expression (RNA-seq) and chromatin accessibility (ATAC-seq) data from NF-LV-CM (blue) and chromatin accessibility (ATAC-seq) and chromatin interaction data from hiPSC-vCM (light blue). Chromatin interaction loops visualize the interaction of CRISPRi target regions (grey) with the interacting promoters HEY2 (**a**) and KCNJ2 (**b**). Only interactions with an interaction strength above the background signal are shown. The line thickness is scaled according to the interaction strength (signal over background), and the color visualizes the statistical significance. ATAC-seq data from NF-LV-CM were previously published¹.

Reviewer figure 2: Chromatin interaction analysis of perturbed domains in hiPSC-aCM

a,b Shown are gene expression (RNA-seq) and chromatin accessibility (ATAC-seq) data from NF-LA-CM (green) and chromatin accessibility (ATAC-seq) and chromatin interaction data from hiPSC-aCM (light green). Chromatin interaction loops visualize the interaction of CRISPRi target regions (grey) with the interacting promoters NPPA and NPPB (a) as well as TBX5 (b). Only interactions with an interaction strength above the background signal are shown. The line thickness is scaled according to the interaction strength (signal over background), and the color visualizes the statistical significance. ATAC-seq data from NF-LA-CM and hiPSC-aCM were previously published^{1,6}.

Reviewer figure 3: Comparative analysis of LV-NF-CM and hiPSC-vCM

a) Fraction of NF-LV-CM PIDs detected in hiPSC-vCM

b) CpG methylation heatmap of PI-CRE detected in NF-LV-CM and flanking regions. Plotted are CpG methylation data for NF-LV-CM and hiPSC-vCM.

c,d) Cumulative relative interactions and number of PIDs per CM marker gene (112 genes, PanglaoDB⁷) detected in NF-LV-CM and hiPSC-vCM. Two-tailed Mann-Whitney test.

5. This might be out of the central scope of this manuscript. But to support the conclusion that “CRISPRi precisely silences individual regulatory elements”, an additional CRISPRi group targeting region 3 should be included in Fig. 8d.

Answer: We targeted region 3 using CRISPRi, and the results confirmed the specificity of the perturbation. These findings are in line with the individual silencing of region 4 and the combinatorial silencing of regions 3+4 depicted in Fig. 8 and with results previously reported by our group⁸. We have updated Fig. 8d as suggested.

Updated Fig. 8d: Functional relevance and promoter interactions of disease-associated non-coding variants in the KCNJ2 locus.

d Chromatin accessibility changes assessed by ATAC-seq after CRISPRi silencing of region 3 and 4 individually or in combination. Control tracks (black) represent corresponding non-targeting CRISPRi conditions. Region 3 was silenced using AAV-CRISPRi, while region 4 and the combined regions 3 and 4 were silenced using lenti-CRISPRi. Shown are merged data from n=3-4 biological replicates.

6. The action potential duration (APD) varies significantly due to difference in ion channel expressions among individual hiPSC-CMs. The sample size (N = 6) in Fig. 8f is too small to draw a strong conclusion. It would be helpful to include CRISPRi target region 1 (promoter), which completely silenced gene expression (Fig.8e), as a positive control group in Fig.8f. In addition, patch clamp recordings using

SyncroPatch 384 might be more sensitive than MEA for detecting APD changes and should be considered.

Answer: As suggested, we increased the sample size and performed MEA recordings following CRISPRi-mediated KCNJ2 promoter silencing. These experiments revealed a significantly prolonged AP duration, consistent with the gene silencing effect and the reduced Kir2.1 currents measured using patch clamp (Fig. 8). Notably, the AP extension observed after KCNJ2 promoter silencing exceeds the AP extension detected after silencing of individual enhancers (region 3, 4), as expected given that CRISPRi of the KCNJ2 promoter resulted in effective (>99%) silencing of KCNJ2 expression.

The MEA technology measures currents from a syncytium of cardiomyocytes in a 2D monolayer, rather than from individual cells. This explains the lower variability of data points obtained from MEA experiments compared to patch clamp measurements, which detect Kir2.1 currents in individual cells.

For the revised manuscript we generated new MEA data, which were analyzed using the "Cardiac Analysis Tool (Axion Biosystems)" that allows for the review and correction of the Q- and T-wave annotations provided by the "AxIS Navigator (Axion)" software. This analysis confirms the previous observations.

In contrast to MEA experiments, which take advantage of spontaneous depolarization events, patch clamp recordings of action potentials require the injection of current. Automated patch clamp can measure ion channel currents accurately but has proven inefficient at capturing action potentials⁹ because of the high access resistance that results in a voltage drop¹⁰. This implies that current-clamp measurements should be interpreted with caution, as both the amplifier circuitry and low seal resistances can necessitate high amounts of injected current and produce artificial readouts of resting membrane potentials, and therefore possibly misreport AP morphology. We therefore believe that additional patch clamp measurements for the detection of AP duration are not of importance for the current manuscript. We want to note that AP data cannot be extracted from the existing patch data generated in this study.

In the revised manuscript, we added new MEA data generated during the second revision, including data for CRISPRi-mediated promoter silencing (region 1), as suggested.

New Fig. 8f: Functional relevance and promoter interactions of disease-associated non-coding variants in the KCNJ2 locus.

f Field potential durations (FPD) were measured in hiPSC-vCM after AAV-mediated CRISPRi silencing of KCNJ2 region 1, 3 and 4 using multi-electrode arrays (MEA). Reported FPD values are corrected using the Fridericia correction formula (FPDc). Data shown from 32-36 biological replicates (mean ± SEM). Kruskal-Wallis test with Dunn's adjustment for multiple comparisons ***P < 0.001 as compared to control.

Minor

1. Please provide justification for why only male cardiac biopsies were used in this study.

Answer: We agree that data from female biopsies would be essential to analyze and highlight sex-dependent mechanisms of chromatin organization, as documented for non-CM cells. However, as stated in the limitations of our study, we were unable to collect a matching cohort of female biopsies. We especially lack suitable age-matched non-failing and non-diseased tissue samples that would allow such analysis.

2. Lin767 states: “hiPSC-vCM were seeded at a density of 30,000 cells in 5 μ l media onto Geltrex-coated recording electrodes of a 48-well CytoView MEA plate (Axion Biosystems) using the spot seeding method. Six hours after seeding, 300 μ l media was added to each well.” It seems unrealistic that 5 μ l of media would adequately support cell viability for 6 hours in a 48-well-plate well. Please clarify this and ensure a reproducible protocol.

Answer: The stated 5 μ l volume is correct and follows manufacturer recommendations for the spot seeding method (5-10 μ l until cells attach). To ensure a humidified atmosphere preventing evaporation of the cell culture medium, PBS was added to the space between the wells/measurement areas. We added this information to the revised method section.

Reviewer #5 (Remarks to the Author):

In this study, the authors investigate chamber-specific cis-regulatory elements (PID-CREs) that interact with their target genes using good quality Hi-C data. They identify PID-CREs that are specific to atrial and ventricular tissues and demonstrate their role in regulating chamber-specific marker genes by CRISPRi. Furthermore, the study highlights an enrichment of SNPs within these regulatory elements that are associated with arrhythmia-related GWAS signals, including those linked to long QT syndrome, thereby providing a valuable framework for investigating and functionally validating disease-associated genetic variants.

The authors have carefully and thoroughly addressed all of Reviewer 1's comments. The revisions prompted by these comments have significantly strengthened the manuscript. Notably, the authors implemented several key improvements that enhance the robustness and functional relevance of their findings:

As recommended by Reviewer 1, the authors performed CRISPR interference (CRISPRi) targeting chamber-specific PID-CREs in atrial and ventricular hiPSC-derived cardiomyocytes (hiPSC-CMs), which led to changes in the expression of target genes.

They validated one of the CRISPRi results by performing complementary CRE deletion experiments, which yielded consistent outcomes.

In addition, the authors incorporated new functional data to support the impact of CRE perturbation in iPSC-CMs. Specifically, they conducted electrophysiological experiments, presented in Figure 8, which functionally validated two CREs at the electrophysiological level.

In their response, the authors also referenced recently published studies that corroborate their findings, particularly regarding the presence and relevance of PID-CREs.

Minor comment:

In their CRISPRi experiments, the authors used atrial and ventricular cardiac differentiation hiPSC. It would be helpful to report the enrichment of atrial and ventricular cardiomyocytes obtained from each respective differentiation, to provide better context for the interpretation of the functional perturbation results.

Answer: Our differentiation protocol yields a high fraction of either ventricular or atrial hiPSC-CM with >90% cTNT+ CM after metabolic selection (new Supplementary Fig. 18). Sub-type specification efficiency was confirmed by flow cytometry using antibodies specific for MLC2a (MYL7, atrial) and MLC2v (MYL2, ventricular) as previously reported¹¹. The atrial differentiation protocol achieved approximately 90% MLC2a+ CM. In comparison, the ventricular differentiation resulted in approximately 90% of MLC2v+ CM, highlighting the validity of our differentiation strategy. In addition, hiPSC-vCM showed a slight increase in MLC2a staining (17.7% MLC2a+/MLC2v+ hiPSC-vCM, new Supplementary Fig. 18). These results are in line with a previous report¹². We further confirmed the quality of each differentiation visually based on the distinct morphology (Reviewer Fig. 4) and characteristic spontaneous beating frequencies of hiPSC-aCM and hiPSC-vCM¹². To verify this quantitatively in an unbiased manner, we performed an RNA-seq analysis after the

*hiPSC-aCM and hiPSC-vCM differentiations used for the comparative analysis in Figs. 4 and 5. Both hiPSC-aCM and hiPSC-vCM display robust and specific expression of atrial and ventricular marker genes using the respective differentiation protocol. This includes the chamber-specific genes studied here. The ventricular CM marker genes *KCNJ2* and *HEY2* are 10- and 67-fold higher expressed in hiPSC-vCM vs. hiPSC-aCM. On the other hand, *TBX5* and *NPPA* show the expected 4-fold higher expression in hiPSC-aCM as compared to hiPSC-vCM. We added this information to the results and method sections, as well as the new Supplementary Fig. 18.*

(Figure legend is provided on the following page)

New Supplementary Fig.18: Characterization of hiPSC-aCM and hiPSC-vCM.

a-d Flow cytometric analysis of marker genes characteristic for CM (cTNT) as well as for atrial (MLC2a, MYL7) and ventricular (MLC2v, MYL2) CM subtypes. Shown are representative data for hiPSC-aCM (green) and hiPSC-vCM (red) stained with fluorescently labeled antibodies and the respective staining controls (grey). The relative cell numbers (%) are stated using the same color code.

e-h RNA-seq gene expression analysis of hiPSC-aCM and hiPSC-vCM. Heatmaps display z-score transformed gene expression data of atrial (**e**) and ventricular (**f**) marker genes. Columns were hierarchically clustered. Bar plots (**g,h**) highlight relative expression data for genes studied in this manuscript. $n=4$ biological replicates, $*P < 0.05$; Two-tailed Mann-Whitney test.

Reviewer figure 4: Morphology and marker expression of hiPSC-aCM and hiPSC-vCM.

Representative pictures of hiPSC-aCM and hiPSC-vCM stained using antibodies targeting the atrial and ventricular markers MLC2a and MLC2v, respectively, and the DNA stain DAPI. (Images were generated by Clarissa Finke, University of Würzburg.)

Overall, I find the authors' responses to be adequate, and the revised manuscript acceptable for publication in its current form. I therefore recommend acceptance.

Answer: Thank you for the positive feedback and for recommending the manuscript for publication. We are especially grateful for reviewing the manuscript to replace the non-responding reviewer 1.

References

- 1 Hocker, J. D. *et al.* Cardiac cell type-specific gene regulatory programs and disease risk association. *Sci Adv* **7** (2021). <https://doi.org/10.1126/sciadv.abf1444>
- 2 Nauffal, V. *et al.* Monogenic and Polygenic Contributions to QTc Prolongation in the Population. *Circulation* **145**, 1524-1533 (2022). <https://doi.org/10.1161/CIRCULATIONAHA.121.057261>
- 3 da Costa-Nunes, J. A. & Noordermeer, D. TADs: Dynamic structures to create stable regulatory functions. *Curr Opin Struct Biol* **81**, 102622 (2023). <https://doi.org/10.1016/j.sbi.2023.102622>
- 4 Wolff, J. *et al.* Galaxy HiCExplorer 3: a web server for reproducible Hi-C, capture Hi-C and single-cell Hi-C data analysis, quality control and visualization. *Nucleic Acids Res* **48**, W177-W184 (2020). <https://doi.org/10.1093/nar/gkaa220>
- 5 Heinz, S. *et al.* Simple combinations of lineage-determining transcription factors prime cis-regulatory elements required for macrophage and B cell identities. *Mol Cell* **38**, 576-589 (2010). <https://doi.org/10.1016/j.molcel.2010.05.004>
- 6 Roselli, C. *et al.* Meta-analysis of genome-wide associations and polygenic risk prediction for atrial fibrillation in more than 180,000 cases. *Nat Genet* **57**, 539-547 (2025). <https://doi.org/10.1038/s41588-024-02072-3>
- 7 Franzen, O., Gan, L. M. & Bjorkegren, J. L. M. PanglaoDB: a web server for exploration of mouse and human single-cell RNA sequencing data. *Database (Oxford)* **2019** (2019). <https://doi.org/10.1093/database/baz046>
- 8 Laurette, P. *et al.* In Vivo Silencing of Regulatory Elements Using a Single AAV-CRISPRi Vector. *Circ Res* **134**, 223-225 (2024). <https://doi.org/10.1161/CIRCRESAHA.123.323854>
- 9 Hayes, H. B. *et al.* Novel method for action potential measurements from intact cardiac monolayers with multiwell microelectrode array technology. *Sci Rep* **9**, 11893 (2019). <https://doi.org/10.1038/s41598-019-48174-5>
- 10 Seibertz, F. & Voigt, N. High-throughput methods for cardiac cellular electrophysiology studies: the road to personalized medicine. *Am J Physiol Heart Circ Physiol* **326**, H938-H949 (2024). <https://doi.org/10.1152/ajpheart.00599.2023>
- 11 Cyganek, L. *et al.* Deep phenotyping of human induced pluripotent stem cell-derived atrial and ventricular cardiomyocytes. *JCI Insight* **3** (2018). <https://doi.org/10.1172/jci.insight.99941>
- 12 Kleinsorge, M. & Cyganek, L. Subtype-Directed Differentiation of Human iPSCs into Atrial and Ventricular Cardiomyocytes. *STAR Protoc* **1**, 100026 (2020). <https://doi.org/10.1016/j.xpro.2020.100026>

Reviewer #2 (Remarks to the Author):

The authors have made significant efforts to address the reviewers' comments. I am satisfied with the answers to my comments and the modifications to the MS.

I have no remaining issues.

Answer: Thanks for the positive feedback on the revision of our manuscript and the confirmation that our additional data addressed your questions.

Reviewer #4 (Remarks to the Author):

The authors have addressed my concerns.

Answer: Thanks for the positive feedback on the revision of our manuscript and the confirmation that our additional data addressed your questions.